# PPP1R35 is a novel centrosomal protein that regulates centriole length in concert with the microcephaly protein RTTN

**Andrew Michael Sydor[1], Etienne Coyaud[2], Cristina Rovelli[1], Estelle Laurent[2], Helen Liu[1], Brian Raught[2,3], Vito Mennella[1,4]***

[1]Cell Biology Program, The Hospital for Sick Children, Toronto, Canada; [2]Princess Margaret Cancer Centre, University Health Network, Toronto, Canada; [3]Department of Medical Biophysics, University of Toronto, Ontario, Canada; [4]Department of Biochemistry, University of Toronto, Ontario, Canada

**Abstract** Centrosome structure, function, and number are finely regulated at the cellular level to ensure normal mammalian development. Here, we characterize PPP1R35 as a novel bona fide centrosomal protein and demonstrate that it is critical for centriole elongation. Using quantitative super-resolution microscopy mapping and live-cell imaging we show that PPP1R35 is a resident centrosomal protein located in the proximal lumen above the cartwheel, a region of the centriole that has eluded detailed characterization. Loss of PPP1R35 function results in decreased centrosome number and shortened centrioles that lack centriolar distal and microtubule wall associated proteins required for centriole elongation. We further demonstrate that PPP1R35 acts downstream of, and forms a complex with, RTTN, a microcephaly protein required for distal centriole elongation. Altogether, our study identifies a novel step in the centriole elongation pathway centered on PPP1R35 and elucidates downstream partners of the microcephaly protein RTTN.

DOI: https://doi.org/10.7554/eLife.37846.001

*For correspondence:
mennellalaboratory@gmail.com

**Competing interests:** The authors declare that no competing interests exist.

## Introduction

The centrosome is a membrane-less organelle whose major role is to organize, orient, and regulate the site of microtubule formation. In somatic dividing cells, the centrosome is critical for ensuring faithful and timely chromosome segregation and establishment of the correct cell division axis, whereas in non-dividing and differentiated cells, it is critical for cellular polarization and cilia formation (*Conduit et al., 2015; Vertii et al., 2016*). Centrosomes are essential for normal human development and health (*Nigg and Holland, 2018*). Loss of function mutations in centrosomal proteins, including components of the centriolar cartwheel, elongation machinery, appendages, and pericentriolar material, are responsible for developmental defects such as primary recessive microcephaly (*Barbelanne and Tsang, 2014*), primordial dwarfism (*Care4Rare Canada Consortium et al., 2015; Rauch et al., 2008; Zheng et al., 2016b*), and ciliopathies (*Reiter and Leroux, 2017*). Defects in centrosome number and structure are a major hallmark of tumorigenesis (*Gönczy, 2015; de Cárcer and Malumbres, 2014; Nigg and Holland, 2018*). Recently, studies in mouse models indicated that centrosome over-duplication concomitant with mutations in *p53* drives tumor formation in the epidermis (*Serçin et al., 2016*) and can drive tumor formation in certain other tissues, even in the absence of concurrent $p53^{-/-}$ mutations (*Levine et al., 2017*). Therefore, it is essential to characterize the critical set of proteins required for centrosome assembly to understand the molecular mechanism of disease and identify therapeutic targets (*Nigg and Holland, 2018*).

**eLife digest** Most animal cells contain a structure called the centrosome, which plays a vital role in helping cells to divide for producing new cells. Early in the cell division process, cells make a copy of their centrosome. Each centrosome includes two cylindrical structures called centrioles encased in a complex web of other proteins. The centrioles must get longer for the duplication process to work correctly, but it is not clear which proteins help the centrioles to elongate.

Previous work suggested that a protein called PPP1R35 might be a centrosome protein. To investigate its role, Sydor et al. performed experiments that reduced the amount of PPP1R35 in cells grown in the laboratory. Cells that contained fewer PPP1R35 proteins also contained fewer centrioles; these centrioles were also shorter and lacked some of the proteins that can elongate them.

Super-resolution microscopy found PPP1R35 in the centre of the centrioles, in a region involved in the early stages of elongation. Sydor et al. also found that PPP1R35 interacts with a protein called RTTN, which is linked to centriole elongation.

RTTN contributes to a condition called microcephaly, which prevents the brain from developing properly and results in individuals having a small head. Future work that builds on the findings presented by Sydor et al. could therefore help researchers to understand the causes of microcephaly in patients.

DOI: https://doi.org/10.7554/eLife.37846.002

Due to its important role in cell and tissue homeostasis, the centrosome is built in a highly-regulated, stepwise manner through the assembly of a multiplicity of protein complexes (*Conduit et al., 2015*; *Mennella et al., 2014*). Significant progress has been made in understanding how centrosome duplication begins in most somatic cells—at the G1/S phase boundary—with the assembly of the cartwheel, a nine-fold symmetrical scaffold made of SAS6, STIL, and CEP135. While SAS6 molecules can undergo remarkable self-assembly in vitro, the kinase Plk4 promotes cartwheel formation and centriole duplication by phosphorylating STIL to favor its interaction with SAS6 (*Vulprecht et al., 2012*; *Lin et al., 2013b*; *Dzhindzhev et al., 2014*; *Arquint and Nigg, 2016*). The initial binding of Plk4 to the centriole is governed by CEP63 (*Brown et al., 2013*), CEP152 (*Brown et al., 2013*; *Kim et al., 2013*; *Sonnen et al., 2013*; *Dzhindzhev et al., 2010*; *Hatch et al., 2010*; *Cizmecioglu et al., 2010*), and CEP192 (*Kim et al., 2013*; *Sonnen et al., 2013*). After cartwheel formation, CPAP, recruited by STIL (*Tang et al., 2011*), aids in the formation of the centriole microtubule wall (*Pelletier et al., 2006*; *Schmidt et al., 2009*) by regulating centriolar microtubule plus-end dynamics (*Basten and Giles, 2013*; *Zheng et al., 2016a*). CEP135 facilitates the stabilization of the centriole structure (*Ohta et al., 2002*; *Basten and Giles, 2013*) but may also play a more direct role in initial cartwheel formation as recombinant *Drosophila* SAS6 and Bld10 (*Drosophila* CEP135 homolog) can self-organize into a nine-fold symmetrical cartwheel structure (*Guichard et al., 2017*).

Once the initial steps of procentriole formation occur, centriole elongation can proceed. However, we have a limited understanding of the essential components required for centriole elongation, which happens between S and G2 phases, and how they are assembled in a stepwise manner. CPAP has been shown to interact with CEP120 (*Lin et al., 2013a*) and SPICE (*Comartin et al., 2013*) in a complex that regulates centriole elongation at the centriolar microtubule wall (*Archinti et al., 2010*; *Lin et al., 2013b*). Centrobin has also been implicated in directly regulating centriolar microtubule elongation (*Lee et al., 2010*; *Zou et al., 2005*) and stability by binding to α-Tubulin (*Gudi et al., 2011*) and by regulating CPAP levels (*Gudi et al., 2015*; *Gudi et al., 2014*). Centrobin is further required to recruit CP110, a protein forming a cap-like structure on the distal end of the centriole that suppresses centriole elongation (*Schmidt et al., 2009*). Proximal to CP110, several proteins localized to the distal luminal end of centrioles such as POC5 (*Azimzadeh et al., 2009*), POC1B (*Venoux et al., 2013*), and OFD1 (*Singla et al., 2010*) have been implicated in promoting the elongation of the centriole's distal region. More recently, additional proteins have been identified, namely CEP295 (*Chang et al., 2016*) and RTTN (*Chen et al., 2017*), which have been proposed to play a scaffolding role in the elongation process by connecting the centriole wall to the luminal

centriolar region. However, it remains unclear if there are components in the lumen of the centriole that stabilize interactions with centriolar wall proteins.

RTTN (rotatin) was originally identified as a protein critical for axial rotation and left-right symmetry specification in mice (*Faisst et al., 2002*). Subsequently, mutations in human RTTN have been shown to cause primary microcephaly and primordial dwarfism (*Kheradmand Kia et al., 2012*; *Grandone et al., 2016*; *Care4Rare Canada Consortium et al., 2015*). Recent reports have shed light on the cellular function of RTTN. The *Drosophila* RTTN homolog, Ana3, was demonstrated to be a centrosomal component critical for maintaining the structural integrity of centrioles (*Stevens et al., 2009*), whereas human RTTN, localized near the centriolar cartwheel, has been shown to be dispensable for initial centriole assembly, but critical for formation of a full-length centriole (*Chen et al., 2017*). It remains unclear what factors are downstream of RTTN and how they promote the elongation and stabilization of the centriole once the cartwheel is formed.

Here, we characterize human PPP1R35, the product of the gene *C7orf47*, which was previously identified in fractions co-purifying with centrosomes in a high-throughput mass spectrometry study (*Jakobsen et al., 2011*). Our study demonstrates that PPP1R35 is a centrosomal component located in the proximal centriolar lumen above the cartwheel. We further demonstrate that PPP1R35 is not important for early centriole assembly but is critical for centriole elongation by impacting the recruitment of the microtubule-binding elongation machinery. In addition, we show that PPP1R35 is downstream of RTTN in the elongation pathway and that they form a protein complex. Altogether, we describe a novel centriolar component essential for centriole formation and identify a new mechanistic step downstream of RTTN in the pathway to reach a fully elongated centriole and functional centrosome.

## Results

### PPP1R35 is stably associated with the centrosome

To examine if PPP1R35 is a bona fide centrosomal protein, we generated a U2OS cell line constitutively expressing GFP-PPP1R35 under the control of a low copy protein expression promoter (*Kim et al., 2011*), integrated into the genome through the Flp-In system. GFP-PPP1R35 showed two main protein populations: one enriched in a diffraction limited spot located in the middle of the cell adjacent to the nucleus, consistent with centrosomal localization, and a cytoplasmic pool (*Figure 1*). To verify that the observed PPP1R35 was located at the centrosomes, we transfected the GFP-PPP1R35 U2OS Flp-In cell line with a vector expressing Centrin 1-mCherry and observed co-localization of Centrin 1-mCherry with GFP-PPP1R35 (*Figure 1a*). To examine the dynamics of PPP1R35 during the cell cycle, we conducted long-term live-cell imaging by spinning disc confocal fluorescence microscopy (*Figure 1b* and *Video 1*). PPP1R35 was found on two centrosomes (grandmother and mother) throughout the entire cell cycle (*Figure 1b*). We observed some cells that have four GFP-PPP1R35 spots prior to mitosis (*Video 1*) and noted that a second GFP-PPP1R35 spot was always present after mitosis, suggesting that PPP1R35 is recruited on daughter centrioles prior to mitosis. To verify that both the mother and daughter centrioles have recruited GFP-PPP1R35, we leveraged the ~1.5 x resolution increase of sub-diffraction live cell imaging. In all G2 cells examined, prior to centrosome separation, two GFP-PPP1R35 spots are resolvable on each of the centrioles (grandmother and mother), confirming that PPP1R35 resides on both the mother and daughter centrioles and must be recruited early in the duplication cycle, in S or early G2 phase (*Figure 1c*). To confirm that GFP-PPP1R35 localization is consistent with the endogenous protein, we imaged U2OS cells labeled with antibodies against PPP1R35 and γ-tubulin by confocal microscopy (*Figure 1d*) and cells labeled with antibodies against PPP1R35 and CETN1 by 3D structured illumination microscopy (3DSIM), and observed co-localization (*Figure 1e*). Since the anti-PPP1R35 antibody showed high background staining we used GFP-PPP1R35 to conduct further studies. To ensure that the GFP tag did not impact the localization of the protein, we examined the morphology of the centrosome by 3DSIM and did not observe a difference between WT and GFP-PPP1R35-expressing U2OS cells (*Figure 1—figure supplement 1a*). Furthermore, we verified that the GFP-PPP1R35 construct did not alter centrosome biogenesis by measuring the total number of CEP152-labeled centrosomes in WT and GFP-PPP1R35-expressing U2OS cells (*Figure 1—figure supplement 1b*). In addition, the GFP-

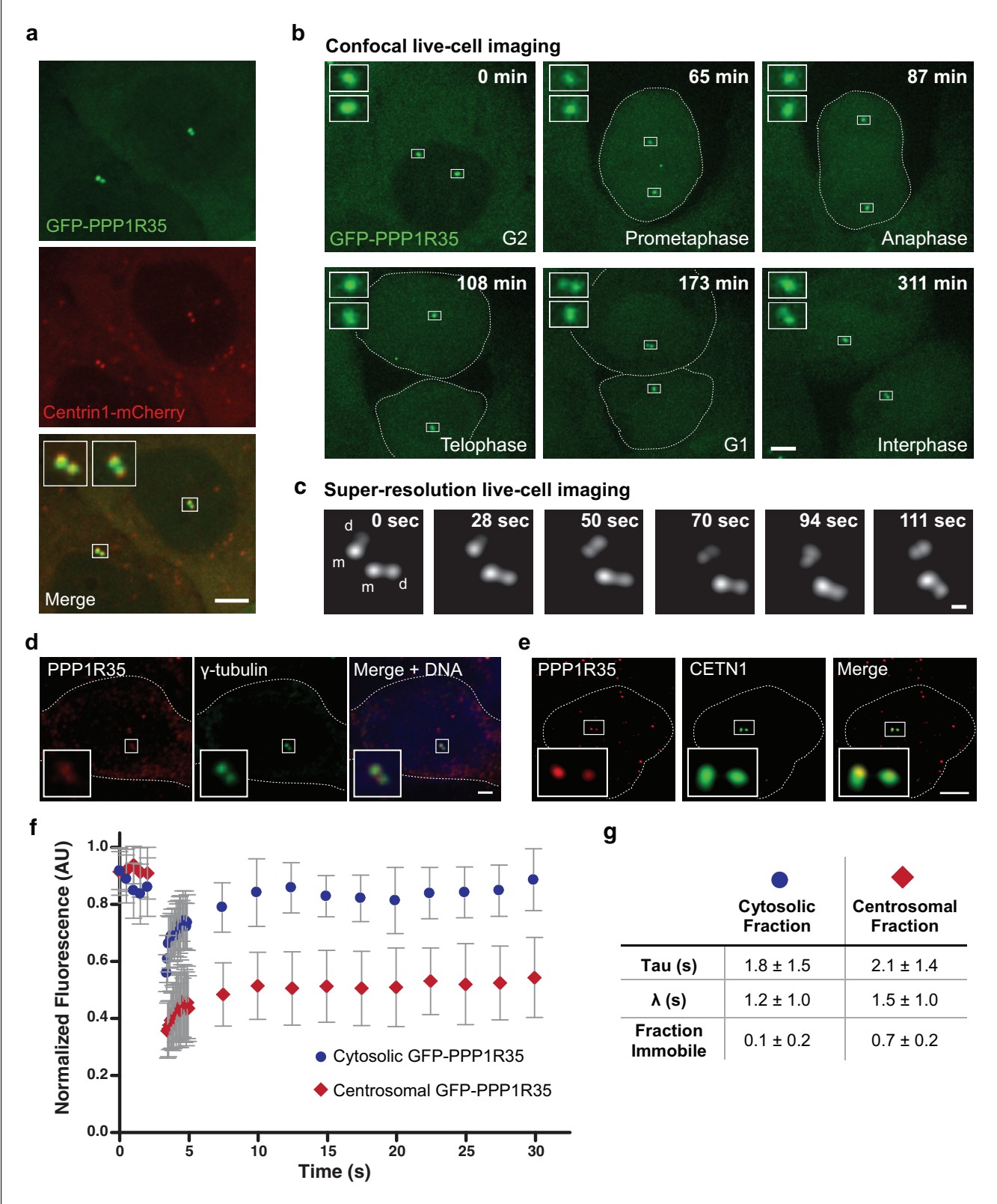

**Figure 1.** PPP1R35 localizes to the centrosome. (a) Confocal microscopy images of live U2OS cells transfected with GFP-PPP1R35 and Centrin1-mCherry. The insets are magnifications of the boxed centrosomes. Scale bar, 5 μm. (b) Still images from time-lapse live-cell spinning-disc confocal microscopy of U2OS cells expressing GFP-PPP1R35. The time elapsed from the start of movie acquisition is printed in each still image. The insets are 5X magnifications of the boxed centrosomes. For clarity, the outlines of mitotic cells are shown. The extra cytosolic green dots observed in the 65 and

*Figure 1 continued on next page*

*Figure 1 continued*

108 min panels are due to camera shot-noise. Scale bar, 5 μm. (**c**) Still images from high-speed live-cell imaging of G1/S or early G2 phase centrosome (assignment based on the engaged nature of the centrosomes) on a Leica HyVolution 2 confocal microscope. The time elapsed from the start of the movie is printed in each still image. The mother and daughter centrioles are indicated by m and d, respectively. Scale bar, 250 nm. (**d**) Confocal microscopy images of fixed U2OS cells labeled with an antibody against PPP1R35 and γ-tubulin. For clarity, the cell outline is shown by a white dotted line. The insets are 4X magnifications of the boxed centrosomes. Scale bar, 2 μm. (**e**) 3D Structured Illumination Microscopy images of U2OS cells stained for PPP1R35 and CETN1 (bottom series). For clarity, the cell outline is shown by a white dotted line. The insets are 6X magnifications of the boxed centrosomes. Scale bar, 2 μm. (**f**) Fluorescence recovery after photobleaching (FRAP) experiments of cytosolic (blue circles) and centrosomal (red diamonds) GFP-PPP1R35. Several images were acquired pre-bleach to establish a baseline and then the respective populations were bleached (at ~3 s) and allowed to recover. Results are a mean of 20 cells and error bars depict the standard deviation. (**g**) Table summarizing the parameters extracted from the FRAP experiments.

DOI: https://doi.org/10.7554/eLife.37846.003

The following source data and figure supplements are available for figure 1:

**Source data 1.** Source data for *Figure 1f and g* (FRAP experiment).
DOI: https://doi.org/10.7554/eLife.37846.006
**Figure supplement 1.** Validation of targeting and biological activity of GFP-PPP1R35.
DOI: https://doi.org/10.7554/eLife.37846.004
**Figure supplement 1—source data 1.** Source data for *Figure 1—figure supplement 1b* (CEP152 recruitment comparison: WT U2OS vs GFP-PPP1R35 U2OS).
DOI: https://doi.org/10.7554/eLife.37846.005

PPP1R35 construct rescues the centriole duplication phenotype when PPP1R35 levels are knocked down by siRNA targeting the 3' untranslated region (3'UTR, see below Figure 3c).

Next, to determine whether PPP1R35 was continuously recruited or was stably associated to the centrosome, we performed Fluorescence Recovery After Photobleaching (FRAP) experiments. Comparison of the fluorescence recovery curves of the cytoplasmic versus centrosomal PPP1R35 pools revealed that centrosomal PPP1R35 did not fully recover to pre-bleach levels after photobleaching, therefore indicating that the protein has low turnover and is stably associated at the centrosome (*Figure 1f,g*). This observation is consistent with a previous analysis that identified PPP1R35 as co-purifying with centrosomal components and observed only a 22% turnover in centrosomal PPP1R35 as measured by stable isotope labeling of amino acids in cell culture (SILAC) mass spectrometry (*Jakobsen et al., 2011*). Altogether, our imaging experiments demonstrate that PPP1R35 is a resident protein of the centrosome and is recruited to the nascent daughter centriole early in the duplication cycle.

## PPP1R35 localizes to the under-characterized proximal centriolar lumen above the cartwheel

To further dissect the role of PPP1R35 at the centrosome, we used super-resolution microscopy to precisely map the position of PPP1R35 relative to several reference markers, whose position at the centrosome has been previously characterized by EM and fluorescence imaging (*Figure 2*). To perform these experiments, we used linear 3DSIM, a technique that provides a 2-fold resolution increase over standard confocal/widefield fluorescence microscopy, which is sufficient to resolve the relative distribution of many centrosomal proteins and allows for straightforward multicolor imaging (*Sydor et al., 2015*; *Mennella et al., 2012*). 3DSIM imaging showed that GFP-PPP1R35 is located in the centrosomal lumen, as suggested by the position of PPP1R35 in the middle of the ring structure formed by CEP152 (*Hatch et al., 2010*; *Cizmecioglu et al., 2010*) (*Figure 2a*). Next, we used several proximal (SAS6, CEP135, CPAP, CEP250) and distal

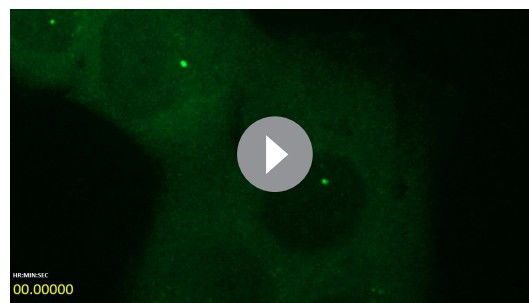

**Video 1.** Live-cell imaging movie of U2OS cells expressing GFP-PPP1R35. Confocal microscopy movie of live U2OS cells stably expressing GFP-PPP1R35.

DOI: https://doi.org/10.7554/eLife.37846.007

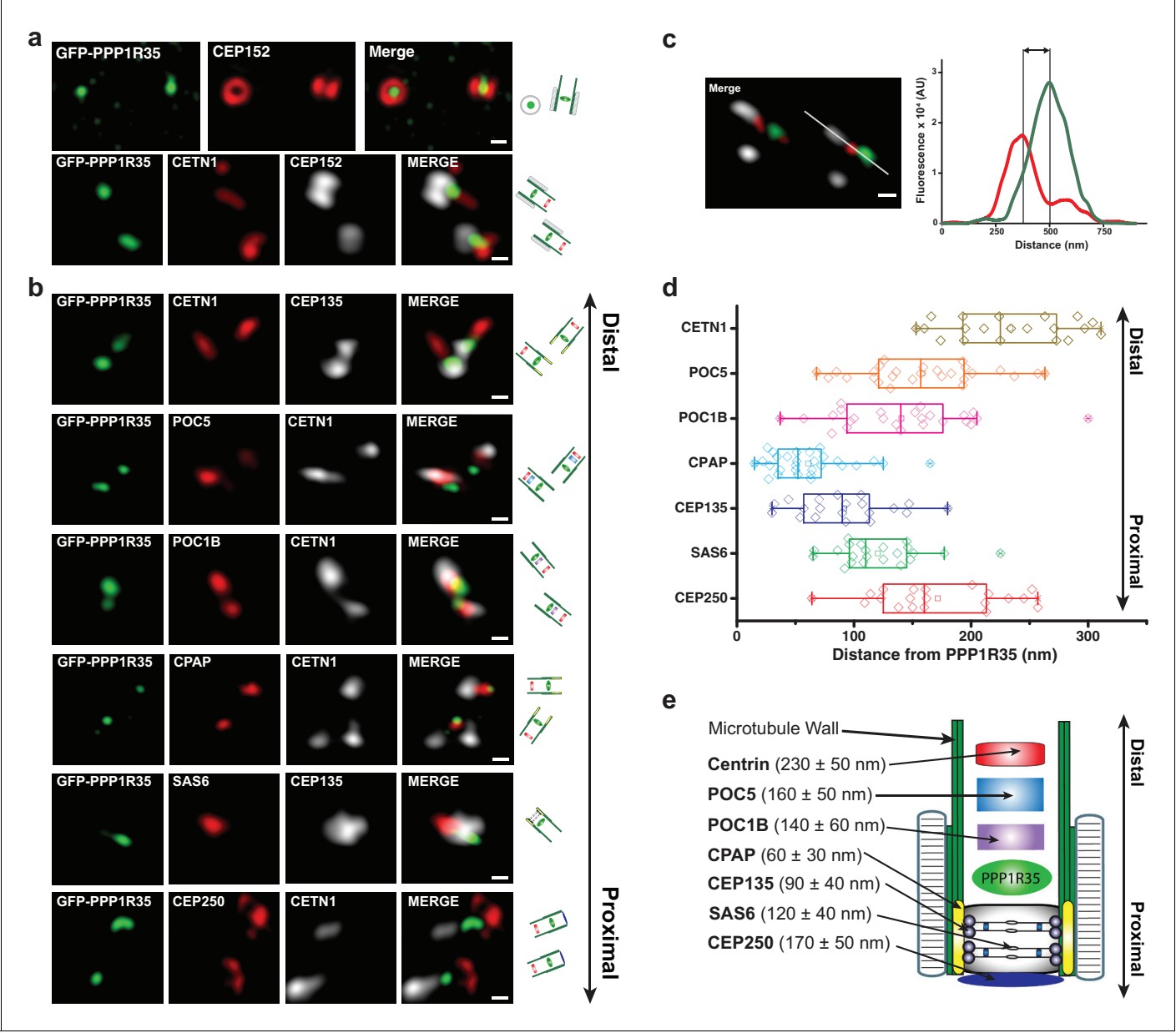

**Figure 2.** PPP1R35 localizes to the proximal end of the centriolar lumen above the cartwheel. (a,b) 3D structured illumination microscopy (3DSIM) micrographs of U2OS cells expressing GFP-PPP1R35 (treated with digitonin to remove the cytoplasmic PPP1R35 population) and co-stained with CEP152 (a) and various other markers (b) labeling the distal (POC5, POC1B, CETN1) and proximal (SAS6, CEP135, CPAP, CEP250) regions of the centriole. The cartoon depiction of the centriole is alongside the images to assist in orienting the 3DSIM micrographs. Scale bars, 250 nm. (c) By measuring the distances between the fluorescence maxima of PPP1R35 and various centriolar markers on the same z-plane, the distance between GFP-PPP1R35 and the corresponding marker was determined. The fluorescence intensity along the line drawn in the micrograph (green, GFP-PPP1R35; red, POC5; white, CETN1) is plotted as a function of the distance along the line. The grey lines are centered at the fluorescence maxima and the distance between these lines corresponds to the distance between GFP-PPP1R35 and the corresponding marker (POC5 in this example). (d) By combining the analysis of the micrographs with these measurements, the position of PPP1R35 was mapped on to the centriole and the distances are shown in a Tukey box and whiskers plot, with the whiskers representing datum within an interquartile range of 1.5 and the band in the box as the mean. (e) A cartoon depiction of the localization of PPP1R35 in the centriole relative to the markers depicted in (b) and (d) with the average distances and standard deviations noted.

DOI: https://doi.org/10.7554/eLife.37846.008

The following source data is available for figure 2:

**Source data 1.** Source data for *Figure 2d and e* (PPP1R35 mapping measurements).

DOI: https://doi.org/10.7554/eLife.37846.009

(CETN1, POC1B, POC5) proteins to locate the position of PPP1R35 along the centrosomal longitudinal axis. Qualitative assessment of the 3DSIM micrographs showed that the position of PPP1R35 is biased toward proximal markers such as CPAP and CEP135 more than either the utmost proximal (i.e. CEP250) or distal (i.e. CETN1) ends of the centriole (*Figure 2b*). To precisely map PPP1R35, we performed a quantitative analysis of the distance between PPP1R35 and many centriolar reference markers. We collected hundreds of 3DSIM images and analyzed micrographs with centriole side views where PPP1R35 was in the same z-plane of the protein of interest for measurement to avoid distortions due to anisotropic resolution (*Figure 2c*). 3DSIM molecular mapping shows that PPP1R35 is located furthest from the distal end proteins (Centrin-1: 230 ± 50 nm; POC5: 160 ± 50 nm; POC1B: 140 ± 60 nm), but closer to proximal end markers such as CEP135 (90 ± 40 nm) and CPAP (60 ± 30 nm), yet not as proximal as SAS6 (120 ± 40 nm) or CEP250 (170 ± 50 nm) (*Figure 2d*). Together, we conclude that PPP1R35 localizes to the proximal centriolar lumen just above the cartwheel (*Figure 2e*).

## PPP1R35 is critical for centriole component recruitment

Since PPP1R35 is recruited early in the centrosome duplication pathway, we hypothesized that it might play a role in regulating centrosome biogenesis. To test this possibility, we depleted PPP1R35 protein levels in U2OS cells by targeting the mRNA with two non-overlapping siRNA strands, one designed to be complementary to an exon in the conserved C-terminal region and the second to the 3' UTR of the PPP1R35 mRNA (*Figure 3a,b*). The specificity of the siRNA strands toward PPP1R35 was validated by western blotting of cells expressing GFP-PPP1R35 (*Figure 3—figure supplement 1*) and RT-qPCR (*Figure 3—figure supplement 2*). We opted to deplete cells of PPP1R35 via siRNA rather than CRISPR/Cas9 gene editing since previous studies demonstrated cell lethality in the absence of PPP1R35 (*Hart et al., 2015*; *Neumann et al., 2010*). Cells were treated with siRNA for 72 hr, thereby allowing cells to progress through multiple cell cycles and accumulate any centriolar defects (*Figure 3b* and *Figure 3—figure supplement 3*). With both siRNA strands, a significant decrease was observed in centrosomal staining of CEP152, a protein recruited in the last stages of daughter centriole formation (*Fu et al., 2016*)(*Figure 3c*). This phenotype is rescued by exogenously expressing GFP-PPP1R35, demonstrating the specificity of the siRNA and the resultant phenotype of PPP1R35 loss (*Figure 3c*). We next sought to narrow down the stage of centrosome duplication at which PPP1R35 plays a role by labeling PPP1R35-depleted cells with several centrosomal proteins sequentially recruited during its assembly (*Conduit et al., 2015*; *Fu et al., 2015*; *Loncarek and Bettencourt-Dias, 2018*). This analysis revealed that centriolar components recruited early in the pathway such as SAS6 (*Dzhindzhev et al., 2014*), CEP135 (*Loncarek and Bettencourt-Dias, 2018*; *Fu et al., 2015*) and Centrin 1 (*Middendorp et al., 1997*), are modestly affected at centrioles in the absence of PPP1R35, as opposed to proteins recruited in later stages, such as CEP295 (*Chang et al., 2016*), POC1B (*Venoux et al., 2013*), and CEP152 (*Loncarek and Bettencourt-Dias, 2018*; *Fu et al., 2015*) that are drastically reduced (*Figure 3e* and *Figure 3—figure supplement 4*). To assay for centrosome function, we examined whether centrosomes could recruit the pericentriolar material or efficiently nucleate microtubules in the absence of PPP1R35 by staining for Cdk5rap2 and γ-tubulin. In both cases, we observed a significant reduction upon PPP1R35 knockdown (*Figure 3—figure supplement 5*).

A more significant impact on centriolar components recruited later in the pathway is more noticeable in a time-course experiment, in which cells are assayed at 24 hr intervals after siRNA treatment (*Figure 3f* and *Figure 3—figure supplement 3*). In these assays, there is little change in the recruitment of early-centriolar components such as SAS6, CETN1, and CEP135 up to the 72 hr time point. In contrast, defective recruitment of other components, such as CPAP and CEP152, is present around the 48 hr time point. When cells treated with PPP1R35 siRNA were stained for CEP152 and SAS6, the proportion of engaged centrosomes with cartwheels was not significantly different (*Figure 3g*), further suggesting that PPP1R35 loss does not influence the early stages of centriole biogenesis. It is also noteworthy that at longer timepoints (>72hr), CETN1 levels drastically decrease suggesting that overall centriole formation is being impacted. Altogether, these results demonstrate that PPP1R35 loss of function results in decreased centrosome number and suggest that PPP1R35 is critical for the recruitment of centriolar components after cartwheel formation.

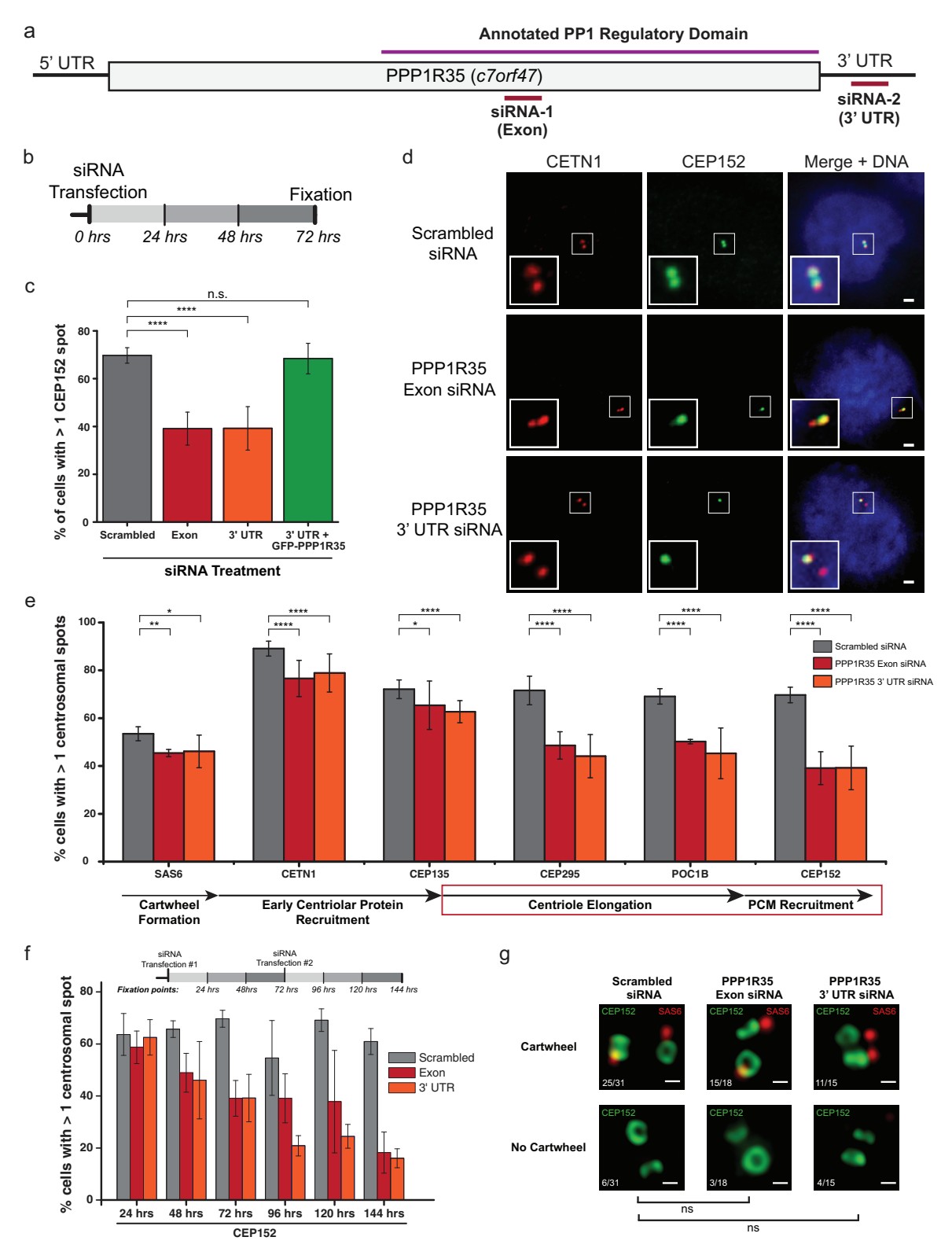

**Figure 3.** PPP1R35 is critical for centrosome duplication. (**a**) Sequence of PPP1R35 with the regions targeted by the siRNA used in this study. (**b**) Schematic representing the timing of the siRNA treatments used in the PPP1R35 knockdown experiments. (**c**) PPP1R35 protein levels were reduced in U2OS cells by treatment with siRNAs targeting either an exon (red bars) or the 3' untranslated region (UTR; orange bars) of the mRNA. The percentage of cells with greater than one CEP152-labeled centrosome, a protein recruited in the later stages of centrosome duplication, is plotted. In this and

*Figure 3 continued on next page*

*Figure 3 continued*

following figures, the grey bar represents U2OS cells treated with a control scrambled siRNA strand; red bar, U2OS cells treated with a siRNA targeting an exon of PPP1R35; orange bar, U2OS cells treated with a siRNA targeting the 3' UTR of endogenous PPP1R35; the green bar represents U2OS cells treated with the 3'UTR-targeting siRNA with exogenously expressed GFP-PPP1R35. (d) Representative micrographs showing decreased numbers of CEP152 positive centrioles in cells treated with PPP1R35 siRNA. Note that CETN1 was not severely perturbed upon treatment with PPP1R35 siRNA. High magnifications of the centrosome for each image are shown in the bottom left corner. Scale bars, 2 μm. (e) U2OS cells treated with either an exon (red bars) or the 3'UTR (orange bars) targeting siRNA (grey bars- control siRNA treated) for the different components of the centriole duplication pathway. The corresponding step of centriole duplication to which the different components belong is depicted with arrows below the plot, with the impacted steps highlighted with a red box. (f) PPP1R35 protein levels were reduced in U2OS cells by treatment with either of the two PPP1R35-targeting siRNA strands, for time points ranging from 24 to 144 hr and subsequently stained for CEP152. The timeline depicted indicate the timing of the siRNA treatments and subsequent time points when cells were fixed and stained for immunofluorescence. (g) 3DSIM micrographs of scrambled- and PPP1R35-siRNA treated cells stained for SAS6 (red) and CEP152 (green). Cells in which two engaged centrosomes were present were analyzed for the presence of a SAS6 cartwheel. Fractions of cells falling into each category are indicated in the top left corner of each micrograph. All scale bars, 250 nm. All error bars show the standard deviation of at least three replicate experiments. Statistics for significance were determined using Barnard's test. All statistical values can be found in *Supplementary file 4*.

DOI: https://doi.org/10.7554/eLife.37846.010

The following source data and figure supplements are available for figure 3:

**Source data 1.** Source data for *Figure 3c and d* (blue tabs) (PPP1R35 siRNA phenotype) and 3f (siRNA time-course, labeled with antibodies against CEP152).

DOI: https://doi.org/10.7554/eLife.37846.019

**Figure supplement 1.** Western blot of siRNA treated GFP-PPP1R35 U2OS cells.

DOI: https://doi.org/10.7554/eLife.37846.011

**Figure supplement 2.** Real-time quantitative PCR analysis of PPP1R35 knockdown by siRNA.

DOI: https://doi.org/10.7554/eLife.37846.012

**Figure supplement 2—source data 1.** Source data for *Figure 3—figure supplement 2* (RT-qPCR).

DOI: https://doi.org/10.7554/eLife.37846.013

**Figure supplement 3.** PPP1R35 siRNA time-course.

DOI: https://doi.org/10.7554/eLife.37846.014

**Figure supplement 3—source data 1.** Source data for *Figure 3—figure supplement 3* (PPP1R35 siRNA timecourse).

DOI: https://doi.org/10.7554/eLife.37846.015

**Figure supplement 4.** Breakdown of centrosome marker counts of cells treated with PPP1R35 siRNA.

DOI: https://doi.org/10.7554/eLife.37846.016

**Figure supplement 5.** Cdk5rap2 and γ-tubulin staining of U2OS cells treated with siRNA targeting PPP1R35.

DOI: https://doi.org/10.7554/eLife.37846.017

**Figure supplement 5—source data 1.** Source data for *Figure 3—figure supplement 5* (PPP1R35 siRNA; cells labeled with antibodies against Cdk5rap2 and γ-tubulin).

DOI: https://doi.org/10.7554/eLife.37846.018

## Biotinylation-dependent proximity mapping of PPP1R35 identifies the microcephaly protein RTTN

To better understand the mechanistic role of PPP1R35 in centriole duplication, we conducted biotinylation-dependent proximity mapping (BioID) (*Roux et al., 2012*) experiments using stable cell lines expressing protein fusions with a FLAG-BirA (R118G) (henceforth referred to as BirA*) tag on either the N- or C-terminus of PPP1R35. BioID analysis revealed a proximity map with several high-confidence hits (FDR score <1%). As expected, the proximity interactome of PPP1R35 shows several centrosomal proteins, including AZI1 (CEP131), CEP85, and KIAA0753 (Moonraker). One of the most robust hits, as evidenced by the high numbers of peptides identified by both N- and C-terminal BirA* tags, is RTTN, a recently characterized protein (H.-Y. *Chen et al., 2017*) whose mutations in patients cause microcephaly (*Care4Rare Canada Consortium et al., 2015*; *Grandone et al., 2016*), dwarfism, and polymicrogyria (*Kheradmand Kia et al., 2012*) (*Figure 4a*; *Supplementary file 1*). Stable HEK293T T-REX Flp-In cell lines showed normal centriolar localization as determined via confocal imaging with the marker CEP152 (*Figure 5f*).

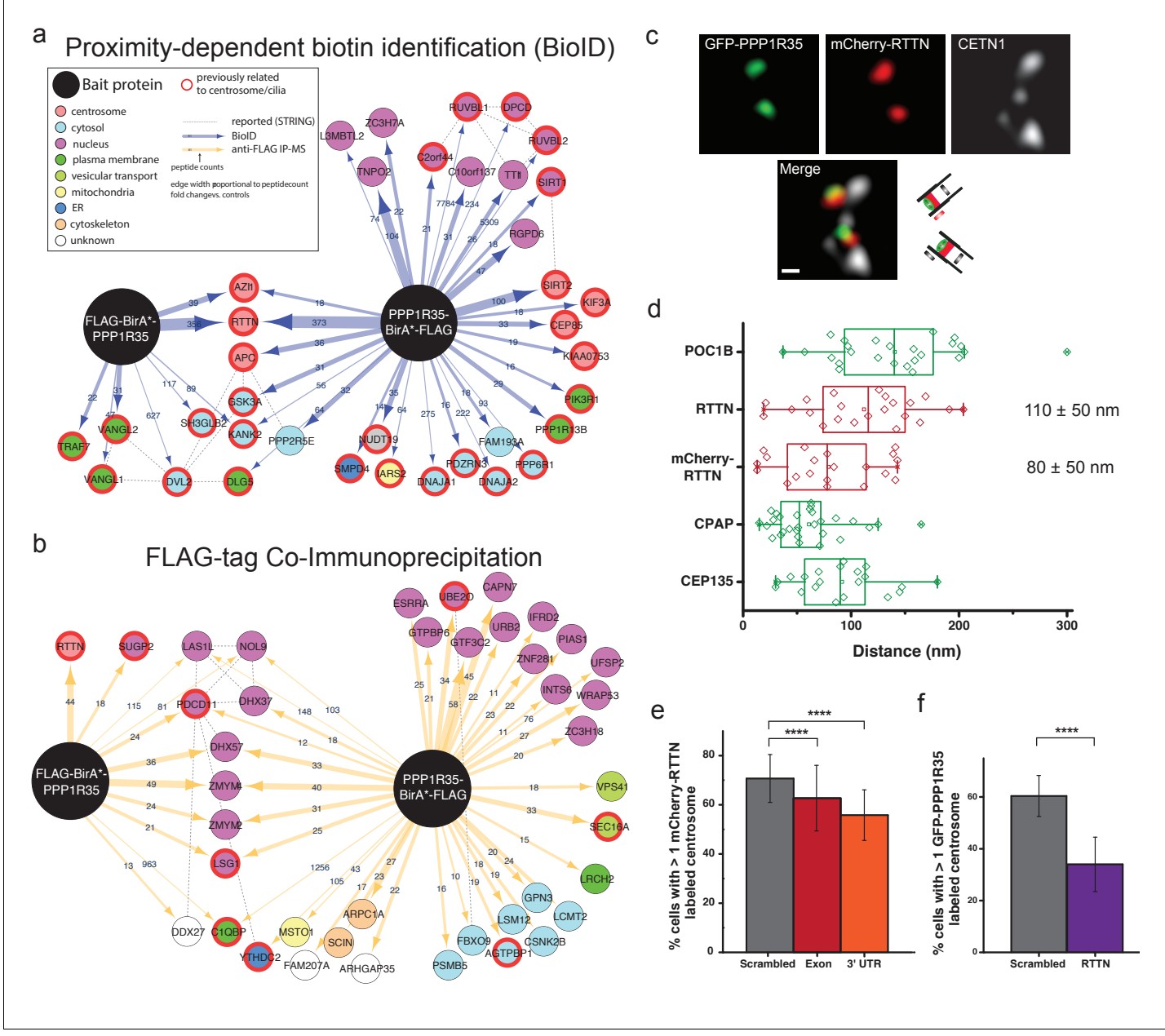

**Figure 4.** PPP1R35 interacts with the microcephaly protein RTTN, which is required for PPP1R35 recruitment. PPP1R35 constructs tagged with a FLAG-tag and mutant R118G Biotin-ligase BirA (BirA*) on either the N- or C-terminus were used for identification of proteins in proximity to PPP1R35 via BioID (**a**) and directly interacting proteins via FLAG-tag immunoprecipitation (**b**), of which the top hits are shown. The arrow thickness indicates number of peptides identified for each hit and the colour of the circle indicates previously annotated sub-cellular localizations. All hits previously related to centrosomes/cilia are indicated by a red outline. (**c**) Localization of mCherry-RTTN relative to centrin-1 and GFP-PPP1R35 in U2OS cells as determined by 3DSIM. Representative cartoons to assist in orienting the images are shown to the right of the merge micrograph. Scale bar, 250 nm. (**d**) 3DSIM mapping of RTTN (via an anti-RTTN antibody and mCherry-RTTN construct) in relation to PPP1R35, conducted using the same methodology as in *Figure 2c* and plotted similarly to *Figure 2d*. The values noted are the average distances and standard deviations. (**e**) U2OS cells expressing mCherry-RTTN were treated with 40 pmol of either a control, scrambled siRNA (grey bars), siRNA targeting the PPP1R35 exon (red bars) or 3'UTR of endogenous PPP1R35 (orange bars). (**f**) U2OS cells expressing GFP-PPP1R35 were treated with 40 pmol control siRNA (grey bars) or siRNA targeting RTTN (purple bars) All error bars depict the standard deviation of at least three experimental replicates. Statistics for significance were determined using Barnard's test. All statistical values can be found in *Supplementary file 4*.
DOI: https://doi.org/10.7554/eLife.37846.020

The following source data and figure supplements are available for figure 4:

*Figure 4 continued on next page*

*Figure 4 continued*

**Source data 1.** Source data for *Figure 4d* (PPP1R35 mapping measurements), 4e (mCherry-RTTN U2OS + PPP1 R35 siRNA), and 4 f (GFP-PPP1R35 U2OS + RTTN siRNA).
DOI: https://doi.org/10.7554/eLife.37846.023
**Figure supplement 1.** Analysis of centrin-positive centrosomes in U2OS cells treated with RTTN siRNA.
DOI: https://doi.org/10.7554/eLife.37846.021
**Figure supplement 1—source data 1.** Source data for *Figure 4—figure supplement 1* (RTTN siRNA labeled with antibody against CETN1).
DOI: https://doi.org/10.7554/eLife.37846.022

## PPP1R35 forms a complex with RTTN and the two proteins are mutually required for each other's recruitment

We reasoned that if PPP1R35 and RTTN are in close proximity to each other and are both located at the centrosome near the cartwheel (H.-Y. *Chen et al., 2017*), they might form a bona fide protein complex. To test this hypothesis, we performed FLAG immunoprecipitation (IP) of the N- and C-terminal tagged PPP1R35 constructs (*Figure 4b* and *Supplementary file 1*). Notably, RTTN was found to form a complex with the N-terminally tagged BirA*-PPP1R35. RTTN was the only protein identified with high confidence by BioID that also co-immunoprecipitated with PPP1R35, suggesting a strong link between this microcephaly protein and PPP1R35 function. Interestingly, IP-mass spectrometry data using the N-terminal FLAG-tagged PPP1R35, but not the C-terminal FLAG-tag construct, detected a high-confidence interaction with RTTN. IP with the PPP1R35 C-terminal FLAG-tag construct still identified RTTN peptide counts above that of the controls, but below our confidence level cut-off (*Supplementary file 1*), indicating that the interaction between the two proteins has been severely impaired but not completely abolished. Altogether, this suggests that the binding site might reside within the conserved C-terminal region of PPP1R35.

RTTN is a 298 kDa protein predicted to have an elongated, solenoid conformation (*Fournier et al., 2013*) that has been recently reported to localize to basal bodies (*Stevens et al., 2009*; *Kheradmand Kia et al., 2012*) and the centrosome (H.-Y. *Chen et al., 2017*; *Stevens et al., 2009*; *Care4Rare Canada Consortium et al., 2015*). Specifically, RTTN has been shown to localize to the proximal lumen of centrioles near CEP135 and the cartwheel (H.-Y. *Chen et al., 2017*). To further characterize the structural and functional relationship of PPP1R35 and the microcephaly protein RTTN, we mapped the position of RTTN relative to PPP1R35 by 3DSIM imaging and quantitative analysis (*Figure 4c*). To detect RTTN we used both an N-terminal mCherry-RTTN construct and an antibody recognizing residues 50–150 of RTTN. Our data show that RTTN localizes to the proximal centriole and it is located in close proximity to PPP1R35, consistent with our BioID findings (PPP1R35 distance from mCherry-RTTN, 80 ± 50 nm; anti-RTTN, 110 ± 50 nm; *Figure 4d*).

To further explore the functional relationship between PPP1R35 and RTTN, we depleted PPP1R35 from U2OS cells by siRNA and examined RTTN recruitment. The presence of RTTN at the centrosome is moderately, yet significantly, diminished upon PPP1R35 depletion (*Figure 4e*). When the reciprocal recruitment was explored by RTTN depletion, a major reduction in centrosomal PPP1R35 was observed (*Figure 4f*). This phenotype appears to be unrelated to the decrease in centriole number expected as a result of RTTN knockdown (*Chen et al., 2017*), because the number of cells with at least 2 centrin spots remains unchanged (*Figure 4—figure supplement 1*), yet the number of cells lacking GFP-PPP1R35 is significantly reduced. These results show that the two proteins co-localize at the centriole and that both proteins are mutually required for each other's recruitment to the centriole, with RTTN exerting a more significant impact on PPP1R35 recruitment to the centriole. Altogether, our data suggest that RTTN and PPP1R35 form a complex and that RTTN acts upstream of PPP1R35.

## Conserved serine phosphorylation sites and the canonical PP1-binding site in PPP1R35 are not critical for centrosome duplication

PPP1R35 is a highly conserved protein whose homologs are found across a wide range of eukaryotic species, ranging from the simple multicellular organism *Trichoplax adhaerens* to *Homo sapiens* (*Figure 5—figure supplement 1*). Interestingly, PPP1R35 homologues are found only in *Holozoa* species, correlating well with species presenting centrosomes, with the exception of *Caenorhabditis*

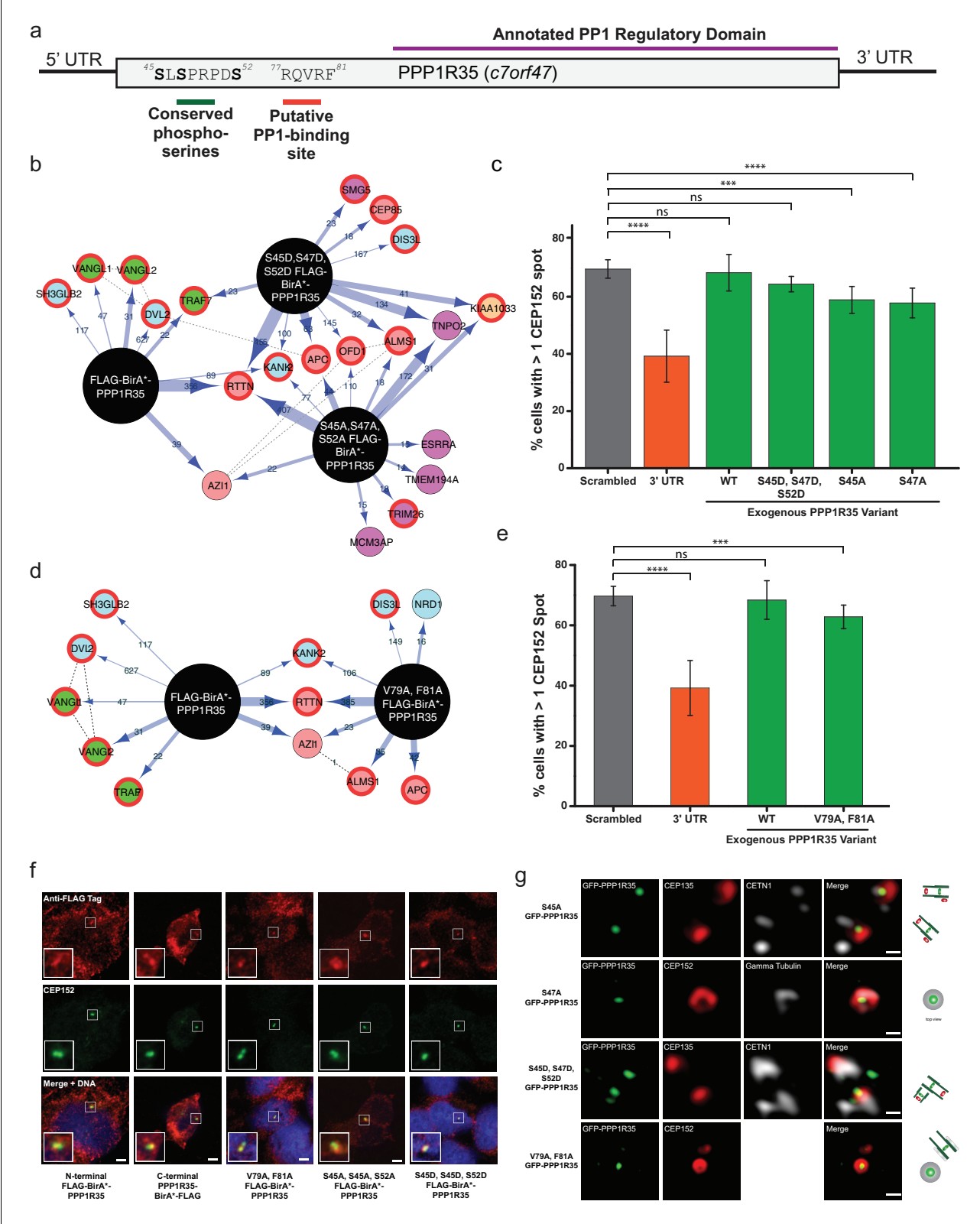

**Figure 5.** Conserved serines and putative PP1-binding site are not critical for centriole formation or interaction with RTTN. (**a**) PPP1R35 contains a series of conserved serine residues that have been previous shown to be Cdk phosphorylation sites and are conserved in mammalian homologs. Furthermore, mammalian PPP1R35 homologs also possess a canonical PP1-binding site. The corresponding residues in both motifs are indicated on the PPP1R35 cartoon. (**b**) BioID results from N-terminal FLAG-BirA*-PPP1R35 mutants carrying non-phosphorylatable (S45A, S47A, S52A) or phospho-mimetic (S45D,

*Figure 5 continued on next page*

*Figure 5 continued*

S47D, S52D) mutations. Please refer to *Figure 4a* for map legend. (c) To probe the role of the conserved serine residues in centriole duplication, amino acids were mutated to either aspartic acids (to mimic the phosphorylated state) or alanine (to mimic the non-phosphorylated state) of the GFP-PPP1R35 construct and tested to see if they could rescue the reduced CEP152-staining phenotype observed upon treatment of U2OS cells with the 3'UTR siRNA. Similar to the triple alanine mutant, we were unable to generate a S52A mutant, suggesting that this residue may be critical for cell viability. (d) BioID results from N-terminal FLAG-BirA*-PPP1R35 mutants carrying the PP1-binding mutation (V79A, F81A). Please refer to *Figure 4a* for map legend. (e) To probe the role of the putative PP1-binding site in centriole duplication, the V79 and F81 were mutated to alanine to abolish PP1-binding in the GFP-PPP1R35 construct and tested to see if they could rescue the reduced CEP152-staining phenotype observed upon treatment of U2OS cells with the 3'UTR siRNA. (f) Localization of HEK293 Flp-In TREX BirA*-tagged PPP1R35 cell lines. Confocal microscopy micrographs of HEK293 Flp-In TREX cells expressing BirA*-tagged WT and mutant PPP1R35 constructs, stained for the FLAG tag (red channel) and CEP152 (green channel). Scale bars, 5 μm. Insets are high magnification of the centrosome. (g) Localization of GFP-PPP1R35 mutants in U2OS Flp-In cells. 3DSIM micrographs of U2OS cells expressing various mutant GFP-PPP1R35 constructs (treated with digitonin to remove the cytoplasmic PPP1R35 population) and co-stained with centriolar markers. The cartoon depiction of the centriole is alongside to assist in orienting the 3DSIM micrographs. Scale bars, 200 nm.
DOI: https://doi.org/10.7554/eLife.37846.024

The following source data and figure supplements are available for figure 5:

**Source data 1.** Source data for *Figure 5c and e* (mutant GFP-PPP1R35 rescue experiments).
DOI: https://doi.org/10.7554/eLife.37846.030
**Figure supplement 1.** Phylogenetic tree of PPP1R35 homologs.
DOI: https://doi.org/10.7554/eLife.37846.025
**Figure supplement 2.** Alignment of selected PPP1R35 homologs.
DOI: https://doi.org/10.7554/eLife.37846.026
**Figure supplement 3.** Summary of high-confidence immunoprecipitation hits for V79A, F81A PPP1R35 and phosphomutants.
DOI: https://doi.org/10.7554/eLife.37846.027
**Figure supplement 4.** siRNA and rescue of HEK293 Flp-In TREX expressing GFP-PPP1R35.
DOI: https://doi.org/10.7554/eLife.37846.028
**Figure supplement 4—source data 1.** Source data for *Figure 5—figure supplement 4* (HEK293 mutant PPP1R35 siRNA).
DOI: https://doi.org/10.7554/eLife.37846.029

*elegans* (*Hodges et al., 2010*). PPP1R35 can be divided into two major domains based on amino acid sequence homology: the highly divergent N-terminal domain and the more conserved C-terminal domain (*Figure 5—figure supplement 2*). Despite its variability across evolution, the N-terminus contains several highly conserved residues in mammalian species including three serine residues (S45, S47, S52 in *Homo sapiens* PPP1R35) previously found phosphorylated in large scale phosphoproteomic studies in both human and mouse cells (*Olsen et al., 2010*; *Dephoure et al., 2008*; *Chi et al., 2008*) (*Figure 5a* and *Figure 5—figure supplement 2*). In particular, S47 and S52 have been reported to be Cdk phosphorylation sites (*Chi et al., 2008*) (*Figure 5a*). As such, we hypothesized that these residues could be candidates for regulating PPP1R35 activity during centrosome duplication, as Cdk2 ensures that centrosome duplication takes place concomitantly with DNA synthesis in S-phase (*Fu et al., 2015*).

To probe the importance of these residues in the interaction with RTTN, we mutated all three serine residues to either non-phosphorylatable alanines (S45A, S47A, S52A) or to phospho-mimetic aspartic acids (S45D, S47D, S52D) and generated inducible HEK293 T-Rex Flp-In cell lines expressing the mutant N-terminal BirA*-PPP1R35 constructs. Neither the triple alanine nor the triple aspartic acid mutant significantly impacted the presence of PPP1R35 at the centrosome, nor its proximity to RTTN (*Figure 5b,g*). Furthermore, co-IP demonstrated that neither phospho-mutant impacted the interaction between PPP1R35 and RTTN (*Figure 5—figure supplement 3*).

We further evaluated the role of PPP1R35 phosphorylation on centriole duplication by examining whether the above phospho-mutants are able to rescue our centriole defect phenotype. When cells were depleted of endogenous PPP1R35 by the 3' UTR-targeting siRNA and expressed the triple aspartic acid mutant (S45D, S47D, S52D) GFP-PPP1R35 in trans, we did not observe any reduction in centrosome number (*Figure 5c*). Despite multiple attempts, we were unable to generate a triple alanine (S45A, S47A, S52A) mutant cell line in U2OS cells, therefore we examined both triple mutant cell lines in HEK293 cells (*Figure 5—figure supplement 4*). Analysis of individual alanine mutants (S45A and S47A) in U2OS cells is also consistent with the notion that that despite their conservation, these resides are not playing a critical for PPP1R35's function in centriole biogenesis (*Figure 5c*).

PPP1R35 is predicted to contain a canonical RVxF PP1-binding site (*Peti et al., 2013*), encompassing residues 77–81 (*Hendrickx et al., 2009*) in the N-terminal domain. This site is conserved only among *Chordata* species (*Figure 5—figure supplement 1* and *Figure 5—figure supplement 2*). Interestingly, disruption of the predicted PP1 binding site by mutating two conserved residues, V79 and F81, to alanine (*Peti et al., 2013*) leads to proper targeting to the centriole (*Figure 5g*) and does not disrupt PPP1R35's proximity to, or interaction with, RTTN (*Figure 5d*, *Figure 5—figure supplement 3*). Furthermore, the V79A, F81A mutant nearly completely rescued our centriole duplication phenotype (*Figure 5e*), suggesting that it is not critical for centriole biogenesis.

## PPP1R35 loss results in shortened centrioles by preventing the recruitment of proteins responsible for centriole elongation

Since PPP1R35 forms a complex with the microcephaly protein RTTN and this protein has been previously linked to centriole elongation, where its loss resulted in shortened centrioles (*Chen et al., 2017*), we investigated whether PPP1R35 knockdown results in diminished centriole length. To this effect, we used 3DSIM to measure the distance between the proximal end of centrioles labeled with acetylated tubulin to CP110, which localizes to the centriole's distal end (*Figure 6a*). Acetylated tubulin has been suggested to be an early tubulin modification during centriole duplication (*Balashova et al., 2009*) and it has been previously used for conducting centriole length measurements (*Chen et al., 2017*). Furthermore, we verified that tubulin acetylation is unaffected when PPP1R35 is knocked down by siRNA (*Figure 6—figure supplement 1*). To ensure that we were examining mature centrioles, we focused our analysis only on mother centrioles in G2 phase cells in which a clear mother and daughter centriole were present. The length of mother centrioles was determined to be 356 ± 65 nm in control-RNAi treated cells, in agreement with previous reports (*Thauvin-Robinet et al., 2014*). When PPP1R35 levels are knocked down by siRNA, we see a significant reduction in centriole length to 263 ± 69 and 246 ± 83 nm for the exon and 3' UTR siRNA, respectively (*Figure 6b*). On the contrary, overexpression of PPP1R35 does not significantly change centriole length (382 ± 89 nm). Due to the small effect observed on RTTN recruitment when PPP1R35 levels are reduced, we hypothesized that the shorter centriole length may be due to the inability of nascent centrioles to recruit proteins involved in elongation. We then examined cells treated with PPP1R35 siRNA for recruitment of proteins involved in either microtubule stabilization/recruitment such as CPAP and SPICE (*Archinti et al., 2010*; *Zheng et al., 2016a*; *Tang et al., 2009*; *Comartin et al., 2013*; *Lin et al., 2013b*) or the elongation of the distal portion of the centriole such as POC5 (*Azimzadeh et al., 2009*) and both proteins were significantly reduced in the absence of PPP1R35. Consistently, CP110, a negative regulator of centriole elongation recruited early in the elongation pathway, was not significantly changed relative to control RNAi-treated cells (*Figure 6c*). Altogether, this demonstrates that PPP1R35 is a critical factor for centriole assembly by promoting recruitment of centriole elongation proteins.

## Discussion

PPP1R35 was initially suggested to be a centrosomal protein by mass spectrometry studies that identified PPP1R35 as co-purifying with isolated centrosomes (*Jakobsen et al., 2011*). Here we show that the uncharacterized protein PPP1R35 is stably associated at the centrosome throughout the cell cycle where it plays a critical role in its elongation. Our loss of function analysis places PPP1R35 relatively early in the centriole duplication pathway, after cartwheel formation and before complete centriole elongation. In addition, we demonstrate that the ultimate downstream effect of PPP1R35 loss is shortened centrioles, suggesting that PPP1R35 directly controls the elongation pathway. It is interesting to note that this diminished centriole duplication defect takes several days to become very pronounced suggesting that either the short centrioles are still competent to duplicate but to a diminished degree, or that only low levels of PPP1R35 are needed for proper biological activity. Phenotypic analysis places PPP1R35 upstream of CPAP, CEP295, SPICE and POC5, which are all proteins involved in centriole elongation (*Tang et al., 2009*; *Lin et al., 2013b*; *Comartin et al., 2013*; *Chang et al., 2016*), but downstream of RTTN, which also affects the recruitment of POC1B, POC5, and CEP295 similarly to PPP1R35 (*Chen et al., 2017*). Furthermore, BioID, IP, and 3DSIM data show that PPP1R35 and RTTN form a protein complex and that this complex is

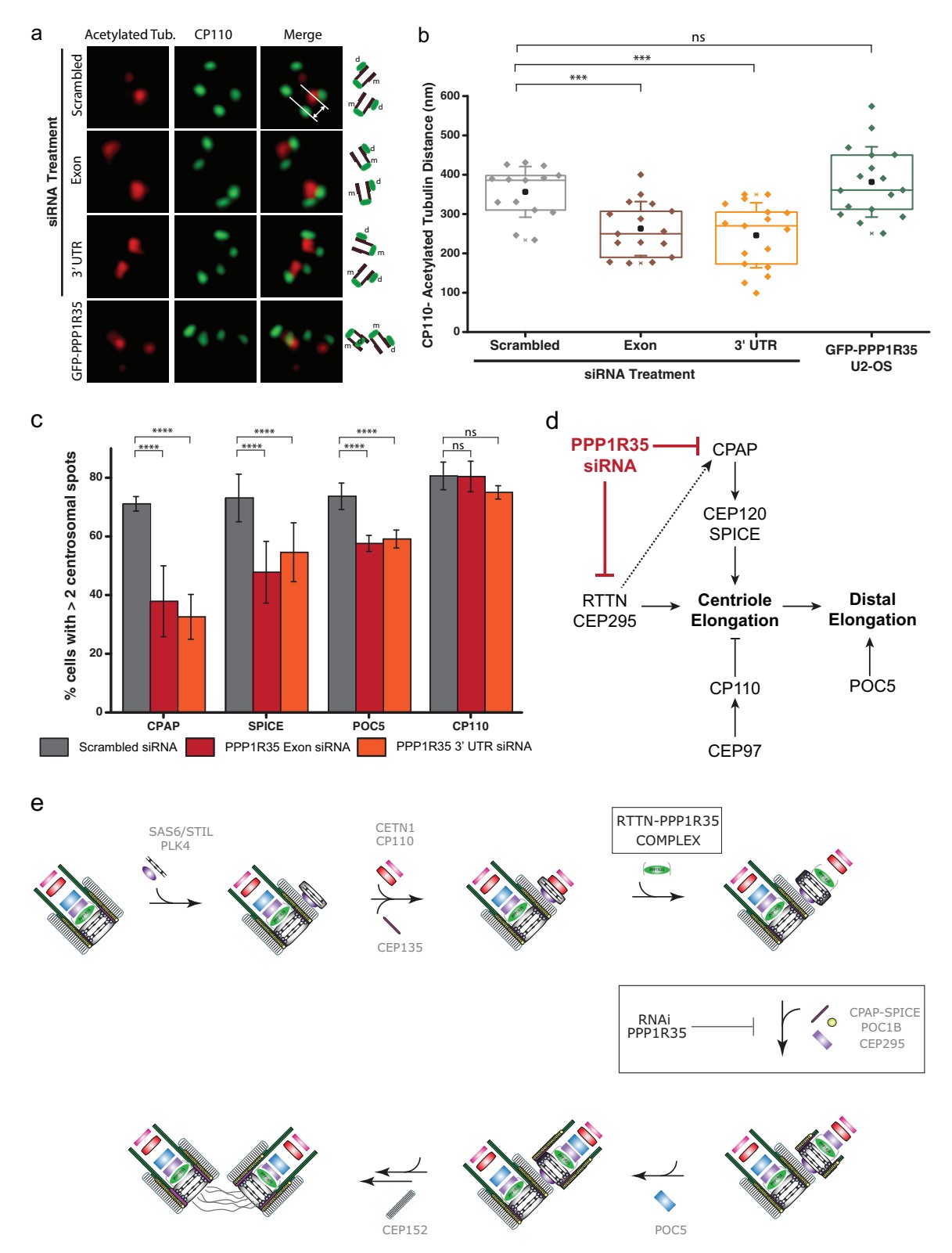

**Figure 6.** PPP1R35 loss results in short centrioles and impacts recruitment of positive-growth centriolar elongation factors. (a) Micrographs of representative centrosomes in U2OS cells stained with CP110 (green) and acetylated tubulin (red) with corresponding cartoon representation. By measuring the distances between the fluorescence maxima of CP110 and the proximal end of the acetylated tubulin signals (designated on the top micrograph by the grey lines) on the same z-plane using 3DSIM, the length of the centrioles was measured. (b) The lengths of centrioles from U2OS

*Figure 6 continued*

cells expressing GFP-PPP1R35 (n = 17) or treated with either control, scrambled siRNA (n = 14) or siRNA targeting an exon (n = 16) or 3'UTR of PPP1R35 (n = 17) for 72 hr plotted as a Tukey box and whisker plot. The average distances and standard deviations are noted in the plot. Statistical significance was determined by a two-tailed T-test. (c) To investigate the centriole elongation phenotype, U2OS cells were depleted of PPP1R35 by siRNA targeting an exon (red bars) or 3'UTR (orange bars) of PPP1R35 (grey bars, control siRNA) and subsequently stained for various proteins involved in centriole elongation. Statistics for significance were determined using Barnard's test. Statistical values can be found in *Supplementary file 4*. (d) Our data demonstrate that PPP1R35 is involved in the CPAP-CEP120-SPICE pathway of centriole elongation and does not impact the CP110-CEP97 pathway, which restricts centriole elongation. (e) Proposed model of PPP1R35 function and mechanism in centriole biogenesis. In this study, we localized PPP1R35 to the proximal lumen of the centriole. By depleting cells of PPP1R35 via siRNA, we identify the step of its recruitment during centriole biogenesis to after cartwheel formation together with the microcephaly protein RTTN. Loss of PPP1R35 results in diminished CPAP, SPICE, POC1B and CEP295 with concomitant loss of downstream components including POC5 and CEP152. The loss of these components, implicated in centriole elongation, results in shortened centrioles.
DOI: https://doi.org/10.7554/eLife.37846.031

The following source data and figure supplements are available for figure 6:

**Source data 1.** Source data for *Figure 6b* (centriole length measurements) and 6c (centriole elongation protein recruitment).
DOI: https://doi.org/10.7554/eLife.37846.034
**Figure supplement 1.** Acetylated tubulin staining of U2OS cells treated with siRNA targeting PPP1R35.
DOI: https://doi.org/10.7554/eLife.37846.032
**Figure supplement 1—source data 1.** Source data for *Figure 6—figure supplement 1* (PPP1R35 siRNA labeled with antibody against acetylated tubulin).
DOI: https://doi.org/10.7554/eLife.37846.033

likely a result of a direct interaction as RTTN is the only protein identified in both BioID and IP experiments.

Whereas several distal luminal proteins have been reported to date (*Azimzadeh et al., 2009*; *Venoux et al., 2013*; *Paoletti et al., 1996*), PPP1R35 is the first to be mapped to the proximal luminal region, an uncharacterized 'outpost' right above the cartwheel. Since most proteins involved in centriole elongation are localized along the microtubule wall or the distal end of the centriole (*Comartin et al., 2013*; *Hatzopoulos et al., 2013*; *Azimzadeh et al., 2009*; *Schmidt et al., 2009*), the interesting distribution of PPP1R35 suggests that it likely acts directly through RTTN, which our imaging show luminal localization for the N-terminus of the protein, in close proximity to PPP1R35.

Our phylogenetic analysis demonstrates that PPP1R35 is conserved in a wide range of species, including *Drosophila*, parasitic worms, and mammals (*Figure 5—figure supplement 1*). PPP1R35 is likely the human homolog of the *Drosophila* protein Reduction of Cnn dots 4 (Rcd4), a protein identified in a large RNAi screen aimed at discovering novel proteins impacting centrosome formation and PCM assembly (*Dobbelaere et al., 2008*). Alignment of *Homo sapiens* PPP1R35 and *Drosophila* Rcd4 results in an overall similarity of 24%, with the greatest homology in the conserved C-terminal domain (*Figure 5—figure supplement 2*). In all identified PPP1R35 homologs, the N-terminus exhibits a large degree of variability, hinting at an organism-specific specialization for this domain. We probed several conserved residues in PPP1R35 including several conserved serines (S45, S47, S52), but none of these mutants appeared to drastically impact centriole biogenesis. Intriguingly, the aforementioned serines and putative PP1-binding site, which are well conserved in *Chordata*, are not conserved in Rcd4. Furthermore, changes to the phosphorylation state did alter the overall BioID proximity map of PPP1R35, including altering the proximity to CEP85, AZI1, and OFD1, the latter two shown to have important roles in ciliogenesis (*Hall et al., 2013*; *Ma and Jarman, 2011*; *Wilkinson et al., 2009*; *Romio et al., 2004*; *Ferrante et al., 2006*).

The centrosomal duplication cycle is closely linked to the cell cycle and tightly controlled by a host of kinases and phosphatases (*Pihan, 2013*; *Fujita et al., 2016*). While kinases inherently possess temporal and spatial specificity, protein phosphatases require a regulatory component to properly function (*Heroes et al., 2013*; *Peti et al., 2013*; *Korrodi-Gregório et al., 2014*). To date, only a handful of centrosomal PP1 regulatory components have been identified (*Katayama et al., 2001*; *Meraldi and Nigg, 2001*; *Mi et al., 2007*; *Helps et al., 2000*; *DeVaul et al., 2013*; *Huang et al., 2005*) and overall knowledge of their interaction and role with PP1 is limited in scope. PPP1R35 is annotated to be a PP1-regulatory protein and contains a canonical PP1-binding site. Despite previous reports that demonstrated PP1-binding and inhibition (*Hendrickx et al., 2009*; *Fardilha et al.,*

*2011*), we were unable to identify any PP1 isoform in our BioID or IP screens using HEK293 cycling cells. This is not completely surprising as previous studies have encountered similar difficulties in identifying interactions between protein phosphatases and their interactors, frequently due to the transient nature of binding (*St-Denis et al., 2016*). We tested the role of this interaction in regard to centriole duplication by mutating the canonical PP1-binding site but found that this interaction with PP1 does not appear to be critical for PPP1R35's role at the centrosome. Overall, this suggests that despite PPP1R35's annotation and previous demonstration as a PP1 regulator, this activity may not be related to centriole biogenesis. However, we cannot yet rule out the possibility that a second, non-canonical PP1-binding site is involved or that the PP1-regulating activity of PPP1R35 is required only in specific cellular functions not investigated here, such as ciliogenesis. Nonetheless, the large number of robust, non-centrosomal BioID hits suggests that PPP1R35 may serve other functions in the cell apart from centriole duplication and perhaps these other functions require PPP1R35's PP1-regulation activity.

Despite the importance of centriole elongation to numerous human diseases, the exact mechanisms through which elongation takes place is still poorly understood (*Loncarek and Bettencourt-Dias, 2018*). To date, two major pathways governing centriole elongation have been described, one positive-growth mechanism acting on assisting microtubule elongation (CPAP/CEP120/SPICE) (*Kohlmaier et al., 2009*; *Schmidt et al., 2009*; *Tang et al., 2009*; *Lin et al., 2013b*; *Comartin et al., 2013*) and a second negative-growth mechanism involving CP110 and CEP97, which form a cap-like structure on the centriole to restrict microtubule growth (*Schmidt et al., 2009*; *Spektor et al., 2007*; *Franz et al., 2013*; *Chen et al., 2002*). However, even with the discovery of additional proteins such as POC5 (*Azimzadeh et al., 2009*), CEP295 (*Chang et al., 2016*), and Centrobin (*Gudi et al., 2015*), all of which impact centriole elongation, we apparently have yet to acquire a complete picture of this process. Here, we have identified a novel key player of this process, PPP1R35. Our data suggest that PPP1R35 primarily impacts the CPAP/CEP120/SPICE and RTTN/CEP295 pathways of centriole elongation (*Figure 6d,e*). Previously, RTTN was proposed to be critical for stabilizing the early procentriole containing STIL, CPAP, and SAS6 and for recruiting CEP295, which in turn can recruit POC5 and POC1B (*Chen et al., 2017*). Our data are consistent with a model where the impact on centriole elongation occurs primarily through PPP1R35's interaction with RTTN. Such a role is consistent with the localization of both PPP1R35 and RTTN, which are uniquely positioned just above the cartwheel in the region where elongation following initial centriole formation would occur. PPP1R35 could be involved in either modulating RTTN's turnover or interactions with other proteins. This reasoning is supported by the observation that RTTN loss has been shown to cause lack of proper CEP295, POC1B, and POC5 recruitment and an interdependency with CPAP and CEP135 (*Chen et al., 2017*), consistent with our observed phenotype when PPP1R35 is knocked down.

Mutations in numerous centriolar proteins have been linked to microcephaly (*Barbelanne and Tsang, 2014*; *Kaindl et al., 2010*), including components involved in centriole elongation such as CPAP (*Leal et al., 2003*) and RTTN (*Care4Rare Canada Consortium et al., 2015*; *Grandone et al., 2016*). The discovery of additional microcephaly proteins will aid in our understanding of the disease and assist in the development of future therapies. Our data show that that PPP1R35 impacts the process of centriole elongation through a close relationship with a known microcephaly protein, RTTN, therefore suggesting that PPP1R35 may be one such candidate microcephaly gene. Future DNA sequencing of microcephaly patients and animal model studies are needed to address this issue.

## Materials and methods

**Key resources table**

| Reagent type (species) or resource | Designation | Source or reference | Identifiers | Additional information |
|---|---|---|---|---|
| Gene (human) | PPP1R35 | MGC clone, TCAG (SickKids) | 4773899 | |
| Gene (human) | RTTN | GE Dharmacon ORFeome | 25914 | |

*Continued on next page*

*Continued*

| Reagent type (species) or resource | Designation | Source or reference | Identifiers | Additional information |
|---|---|---|---|---|
| Cell line (human) | U2OS | ATCC | HTB-96, RRID: CVCL_0042 | |
| Cell line (human) | HEK293 TREX Flp-In | Thermo Fisher Scientific | R78007, RRID: CVCL_U427 | |
| Cell line (human) | U2OS Flp-In | Trimble Lab/ SickKids | | Generous gift from Trimble Lab |
| Antibody | GFP | Abcam | ab13970, RRID: AB_300798 | 1:2000 (IF) |
| Antibody | mCherry | Life Technologies | M11217, RRID: AB_2536611 | 1:200 (IF) |
| Antibody | CETN1 | Millipore | 04–1624, RRID: AB_10563501 | 1:200 (IF) |
| Antibody | CEP152 | Bethyl | A302-480A, RRID: AB_1966084 | 1:500 (IF) |
| Antibody | CEP135 | Bethyl | A302-250A, RRID: AB_1730796 | 1:500 (IF) |
| Antibody | CPAP | ProteinTech | 11517–1-AP, RRID: AB_2244605 | 1:50 (IF) |
| Antibody | SAS6 | Santa Cruz | sc-81431, RRID: AB_1128357 | 1:200 (IF) |
| Antibody | POC5 | Bethyl | A303-341A, RRID: AB_10971172 | 1:500 (IF) |
| Antibody | POC1B | Invitrogen | PA5-24495, RRID: AB_2541995 | 1:50 (IF) |
| Antibody | Gamma-Tubulin | Sigma | T6557, RRID: AB_477584 | 1:5000 (IF) |
| Antibody | CEP250 | ProteinTech | 14498–1-AP, RRID: AB_2076918 | 1:50 (IF) |
| Antibody | CEP295 | Abcam | ab122490, RRID: AB_11129892 | 1:100 (IF) |
| Antibody | RTTN | Abcam | ab113541 | 1:50 (IF) |
| Antibody | Polyglutamylated tubulin | AdipoGen | AG-20B-0020-C100 | 1:400 (IF) |
| Antibody | CP110 | ProteinTech | 12780–1-AP, RRID: AB_10638480 | 1:200 (IF) |
| Antibody | SPICE | Atlas | HPA064843, RRID: AB_2685367 | 1:500 (IF) |
| Antibody | GAPDH | Abcam | ab181602, RRID: AB_2630358 | 1:500 (IF); 1:1000 (Western Blot) |
| Antibody | GAPDH | Millipore | MAB374, RRID: AB_2107445 | 1:200 (IF) |
| Antibody | GFP (for Westerns) | Life Technologies | A11122, RRID: AB_221569 | 1:2000 (Western Blot) |
| Antibody | FLAG-tag | Sigma | F3165-1MG | 1:500 (IF) |
| Antibody | Anti-rabbit HRP (for Westerns) | Cell Signalling | 7074S, RRID: AB_2099233 | 1:2000 (Western Blot) |
| Antibody | PPP1R35 | ProteinTech | 24214–1-AP | 1:50 (IF) |
| Antibody | Acetylated tubulin | Sigma | T7451, RRID: AB_609894 | 1:400 (IF) |
| Antibody | γ-tubulin | Sigma | T6557, RRID: AB_477584 | 1:5000 (IF) |
| Recombinant DNA reagent | pIRES Centrin 1 mCherry | Addgene | 64338 | Deposited by Matthieu Piel |
| Recombinant DNA reagent | pcDNA5-FRT-To-Sept2p | DOI: 10.1083/jcb.201106131 | | Created by Moshe Kim |

*Continued on next page*

*Continued*

| Reagent type (species) or resource | Designation | Source or reference | Identifiers | Additional information |
|---|---|---|---|---|
| Recombinant DNA reagent | pcDNA5/FLAG/TO-FLAG-BirA* | 10.1074/mcp.M114.045658 | | Created by the Raught Lab |
| Recombinant DNA reagent | pcDNA5/FLAG/TO-BirA*-FLAG | 10.1074/mcp.M114.045658 | | Created by the Raught Lab |
| Recombinant DNA reagent | pOG44 | Thermo Fisher Scientific | V600520 | |
| Sequence-based reagent | Scrambled siRNA | Ambion (Life Technologies) | 4390844 | |
| Sequence-based reagent | GAPDH siRNA | Ambion (Life Technologies) | 4390850 | |
| Sequence-based reagent | PPP1R35, exon siRNA | Ambion (Life Technologies) | s48124 | |
| Sequence-based reagent | PPP1R35, 3' UTR siRNA | Ambion (Life Technologies) | s195859 | |
| Sequence-based reagent | RTTN siRNA | Ambion (Life Technologies) | RTTNHSS119506 | |
| Commercial assay or kit | PureLink PCR Purification Kit | Thermo Fisher Scientific | K310001 | |
| Commercial assay or kit | DNA Gel Extraction Kit | Qiagen | 28704 | |
| Commercial assay or kit | Gibson Assembly Kit0 | New England Biolabs | E2611S | |
| Commercial assay or kit | QuikChange II XL Lightning Mutagenesis Kit | Agilent | 200523 | |
| Commercial assay or kit | Mycoplasma Detection Kit | Invitrogen | M7006 | |
| Commercial assay or kit | JetPrime Transfection Reagent | Polyplus | 114–07 | |
| Commercial assay or kit | Lipofectamine RNAiMax | Invitrogen | 13778150 | |
| Commercial assay or kit | GeneJet RNA Purification Kit | Thermo Fisher Scientific | K0731 | |
| Commercial assay or kit | RapidOut DNA Removal Kit | Thermo Fisher Scientific | K2981 | |
| Commercial assay or kit | iScript cDNA Synthesis Kit | BioRad | 1708890 | |
| Commercial assay or kit | SsoAdvanced Universal SYBR Green Supermix | BioRad | 1725271 | |
| Software, algorithm | Barnard's Test (R Package) | Kamil Erguler | https://github.com/kerguler/Barnard | |

## Plasmids and molecular biology

Tables detailing primers (*Supplementary file 2*) and siRNA strands (*Supplementary file 3*) used in this study are available in the Supplemental Material section. The construct pIRES Centrin1 mCherry was a gift from Matthieu Piel (Addgene plasmid # 64338). For cloning experiments, PCR products were amplified from plasmid cDNA (PPP1R35 cDNA, MGC clone Image ID 4773899; RTTN cDNA, GE Dharmacon ORFeome cDNA 25914), verified for specificity of amplification on an agarose gel and purified using the PureLink PCR Purification kit (Thermo Fisher Scientific) or gel-extracted (Qiagen DNA Gel Extraction Kit) when necessary. All cloning experiments were conducted using Gibson Assembly (New England Biolabs) according to the manufacturer's instructions. Site-directed mutagenesis for the single serine mutants was conducted using QuikChangeII XL Lightning site-directed mutagenesis kit (Agilent). Gene synthesis (Invitrogen GeneArt) was used to generate mutant

PPP1R35 containing the triple Ala, Asp, and V79A, F81A mutations and were subsequently cloned into the GFP-containing FRT vector via Gibson Assembly as described above. Newly generated plasmid constructs were verified using Sanger Sequencing (ACGT Corp., Toronto).

## Cell culture

U2OS and U2OS Flp-In cells were cultured in McCoy's 5A medium (Gibco) supplemented with 10% fetal bovine serum (Wisent) and 1 X antibiotic/antimycotic (Gibco, 100 units/ml penicillin, 100 µg/ml streptomycin, and 0.25 µg/ml amphotericin B). HEK 293T TREX cells were maintained in Dulbecco's Modified Eagle's Medium (Gibco) supplemented with 10% tetracycline-free fetal bovine serum (Wisent) and 1 X antibiotic/antimycotic. All cells were cultured at 37°C with 5% $CO_2$ and routinely passed. The cells were tested routinely for mycoplasma contamination using Invitrogen's Mycoplasm Detection Kit.

For cellular transfection of DNA plasmids, JetPrime (Polyplus) was used according to the manufacturer's instructions. Cells were subsequently selected in the appropriate antibiotic (puromycin for constructs in U2OS Flp-In cells and hygromycin for constructs in HEK293 Flp-In TREX cells) to generate cell lines with stably integrated transgenes. For siRNA transfections, Lipofectamine RNAiMax (Invitrogen) was used according to the manufacturer's instructions. All siRNA strands were transfected at a final concentration of 40 nM and cells were assayed 72 hr post transfection. Scrambled siRNA and siRNA targeting GAPDH were used as negative and positive controls, respectively. GAPDH control knockdown efficacy was monitored by immunofluorescence and by western blot. The number of labeled centrosomes per cell (classified into categories as either 0, 1, 2 or >2 labeled centrosome; see *Figure 3d* for example of centrosome spots imaged in cells and *Figure 3—figure supplement 4* for example of categories of centrosomal counts) were manually counted and all siRNA experiments were conducted in at least triplicate except for the HEK293 siRNA experiments that were conducted in at least duplicate.

## Live-cell imaging

For live-cell imaging, cells were seeded on KOH-washed coverglass (Electron Microscope Sciences) to reduce background fluorescence and subsequently left overnight to adhere. The standard culture media was replaced with DMEM medium lacking phenol red (Gibco) supplemented with 10% fetal bovine serum and 1 X antibiotic/antimycotic. Cells were imaged on a Zeiss Axio Observer Spinning-disc microscope equipped with Yogokawa spinning disk head, Phototronics EM CCD camera, and a 63x objective (NA = 1.4). The samples were maintained at 37°C with 5% $CO_2$ during imaging in an incubation chamber. Automated acquisition of a 30 µm z-stack with a 0.75 µm step size every 40–45 min was obtained using the Zeiss Zen Blue software. During acquisition the lowest possible minimal laser power was used to avoid phototoxicity, resulting in movies of an average length of 14 hr.

## Fluorescence recovery after photobleaching (FRAP) experiments

The cells were imaged on a Leica SP8 Confocal DMI6000 microscope equipped with a HyD detector and a 63x (NA = 1.4) oil objective. The samples were maintained at 37°C with 5% $CO_2$ during imaging in an incubation chamber. Samples were bleached with a white light laser for approximately 1.5 s and the subsequent recovery monitored for an additional 26.5 s. The resultant plots were analyzed and fit to a single exponential curve using the build-in FRAP analysis function in the Leica Analysis Suite X software.

## Immunofluorescence

A table detailing all antibodies used in this study, including concentrations and suppliers, is available in the key resources table. Cells were plated onto coverslips (Electron Microscope Sciences; previously cleaned with KOH) and left overnight to adhere. The cells were treated with 0.02% w/v digitonin in PBS for 5 min at RT to remove the cytoplasmic population of PPP1R35 followed by fixation with −20°C methanol for 20 min. The cells were blocked for 1 hr using 5% FBS in PBS supplemented with 0.5% Tween-20. The cells were incubated with primary and secondary antibodies for 40 min each at RT. To detect specific primary antibodies, Alexa 488-, Alexa 568-, or Alexa 647-conjugated IgGs were used as secondary antibodies at a dilution of 1:1000 (Invitrogen). Cell nuclei were stained

with Hoescht 33342 (Thermo Fisher). Cells were mounted with 0.5% n-propyl gallate in 80% glycerol mounting media.

## 3D structured illumination microscopy

3DSIM data were collected using an ELYRA PS.1 (Carl Zeiss Microscopy) with a Plan-Apochromat 63x or 100x/1.4 Oil immersion objective lens with an additional 1.6x optovar. An Andor iXon 885 EMCCD camera was used to acquire images with 101 nm/slice z-stack intervals over a 5–10 µm thickness. The fluorophores were excited with 405, 488, 555 and 647 nm wavelengths and band-pass 420–480, 495–550, 570–620, long-pass 655 and 750 nm filters were used to collect the emission wavelengths. Laser powers at the objective focal plane of 52.6 mW in the 2–12% range, exposure time between 50–250 ms and EMCCD camera gain values between 5 and 50 were used during image acquisition. For each image field, grid excitation patterns were collected for five phases and three rotation angles (−75°; −15°, +45°). The raw data were reconstructed using the SIM module of ZEN Black Software (version 8.1) with noise filter values between −6 and −3. Channel alignment was conducted using calibrated file generated from super-resolution Tetraspec beads (Carl Zeiss Microscopy). If appropriate, whole-volume images or maximum intensity projections were exported as tiff files to be further analyzed in ImageJ/Fiji (NIH).

## Protein mapping and centriole length measurements by 3DSIM

To measure the position of PPP1R35 relative to various centriolar markers, only 3DSIM images in which both the fluorescence maxima of PPP1R35 and the corresponding reference protein were on the same z-slice were analyzed. The distance between the peak maxima for the two markers were determined using the caliper function built in to the Zeiss Zen Black software (see *Figure 2c* for an example). The centriole length measurements were conducted in an identical manner using CP110 as a distal end marker and the acetylated tubulin signal as a proximal end marker (see *Figure 6a* for an example).

## Proximity-dependent biotinylation

BioID was conducted as previously described (*Firat-Karalar and Stearns, 2015*; *Gupta et al., 2015*). To generate stable cell lines expressing recombinant BirA fusion proteins for BioID experiments, HEK293 Flp-In T-Rex cells were co-transfected with the pcDNA5/FRT/TO PPP1R35-FLAG-BirA* or pcDNA5/FRT/TO FLAG-BirA*-PPP1R35 plasmid and Flp Recombinase Expression plasmid pOG44 in a 1:20 ratio, and then selected for multiple passages with increasing antibiotics concentrations to reach final concentrations of 400 µg/ml Hygromycin B (Invitrogen) and 15 µg/ml Blasticidin (Gibco, Thermo Fisher Scientific). HEK293 TREX Flp-In cells expressing the appropriate transgene were cultured until 90–100% confluency and treated for 24 hr with 1 µg/ml tetracycline to induce BirA expression and 50 µM biotin to allow biotinylation of proteins. HEK293T TREX Flp-In cells transfected with a vector containing either the N- or C-terminal FLAG-BirA* but no PPP1R35, were processed in parallel as controls.

Cells were collected, pelleted, and washed three times with PBS prior to freezing. Cell pellets were processed for Bio-ID and FLAG ImmunoPrecipitation (IP) experiments as described previously (*Coyaud et al., 2015*). Interactor classification: bona fide interactors were defined as high confidence protein identifications (ProteinProphet p>0.85) with a SAINT score ≥0.75, based on 4 independent MS runs. Histone hits were eliminated. Fold-change was calculated as described previously (*Coyaud et al., 2015*).

## Western blot

Total cell lysates were collected using RIPA lysis buffer (Pierce) supplemented with mammalian protease inhibitor (BioBasic; 100 mM PMSF, 1 mM Bestatin, 1.5 mM Pepstatin A, 1.4 mM E-64, 0.08 mM Aprotinin, 1 mM Leupeptin) and cell debris pelleted by spinning for 30 min at 12,000 rpm. Protein concentrations were determined using a BCA protein assay kit (Pierce). Protein lysate containing ~15–30 µg of total protein was loaded onto well of 4–12% Bis-Tris gels (Invitrogen). Proteins were transferred to nitrocellulose membrane for 2 hr on an Invitrogen Bolt Minigel Apparatus at 10 V and blocked with 5% skim milk for 1 hr. Membranes were subsequently incubated with

specific antibodies overnight at 4°C. Secondaries conjugated with HRP (Cell Signalling) were used at a 1:2000 dilution. Blots were developed using the ECL Chemiluminescent Substrate Kit (Invitrogen).

## Real-time quantitative polymerase chain reaction (RT-qPCR)

RNA was extracted from cells using the GeneJet RNA Purification kit (Thermo Scientific) and subsequently treated with the RapidOut DNA Removal kit (Thermo Scientific). Purified RNA was quantitated and only RNA with an A260/A280 ratio greater than 1.8 was used for reverse transcription with the BioRad iScript cDNA Synthesis kit with 1 μg of RNA as the template. All quantitative PCR was performed using a CFX Connect Real-Time System (BioRad) with SsoAdvanced Universal SYBR Green Supermix (BioRad) and 500 nM combined primer concentration per well. The relative expression of the target genes were normalized to RNA polymerase II and TATA binding protein transcript levels for each condition and then relative to expression in the scrambled siRNA-treated sample. Primer sequences can be found in *Supplementary file 2*. No-template and no-reverse transcriptase controls were run for each primer pair to confirm the lack of primer–dimer formation/DNA contamination and genomic DNA contamination, respectively. At least three biological replicates were run per condition. Data were analyzed using the CFX Maestro software (BioRad). All kits were conducted as per the manufacturer's protocol.

## Statistical analysis

All siRNA experiments were analyzed as $2 \times 2$ contingency tables in which all cells for a given population (i.e. cells with >1 CEP152 spots) were pooled for all replicates. To determine the p-values compared to the scrambled siRNA control for each dataset, Barnard's Test was used in R with unpooled variances (package by Kamil Erguler; available at https://github.com/kerguler/Barnard) (*Erguler, 2015*). A summary of all statistics for the siRNA experiments can be found in *Supplementary file 4*. For all other statistical tests, the Student's T-Test was used. Error bars represent the standard deviation for all replicates. For all figures, the following conventions were used: ns ($p>0.05$), * ($p\leq0.05$), ** ($p\leq0.01$), *** ($p\leq0.001$), **** ($p\leq0.0001$).

## Phylogenetics

The NCBI protein database was queried with the search term 'PPP1R35' and all resultant hits were downloaded. For species with multiple annotated isoforms, the longest was selected. Any entries that were also annotated as a protein of known function (i.e. transposase, helicase, etc) were removed. Furthermore, only one organism per genus was selected to ensure broad coverage yet avoiding artifacts caused by over-sampled genera. All entries were from the *Holozoa* group of Eukaryotes. In order to ensure that no sequences from other major eukaryotic groups were missed, Delta Blastp searches using both the *Homo sapien* and *Drosophila melanogaster* PPP1R35 sequences were used to search for homologs in representative genera from the remaining eukaryotic groups (exact genera probed are those found in *Figure 1* of Ref. [*Hodges et al., 2010*]). No additional homologs were identified outside of the *Holozoa*. Multiple sequence alignments were performed using Clustal Omega (*Sievers et al., 2011*) with the default settings. The phylogeny was inferred using the Bayesian method implemented with MrBayes v. 3.2.6 (mixed amino acid rate mode) and run for 2.5 million generations until the standard deviation of split frequencies was 0.199. *Drosophila melanogaster* Sds22, a PP1 regulator protein identified to have diverged early from homologous PP1 regulators (*Ceulemans et al., 2002*), was used as the outgroup. Trees were drawn using FigTree v. 1.4.3.

## Acknowledgements

We would like to thank the Trimble and Pelletier labs for vectors and reagents, Dr. Moshe Kim for the empty pcDNA5-FRT-TO- GFP Sept2p plasmid, the Mennella laboratory for insightful discussions and feedback, the National Science and Engineering Research Council of Canada for generous funding. Andrew Sydor is a Restracomp Fellow from the Hospital for Sick Children.

# Additional information

## Funding

| Funder | Grant reference number | Author |
|---|---|---|
| National Science and Engineering Research Council of Canada | Discovery grant, RGPIN-2015-04795 | Vito Mennella |
| The Hospital for Sick Children | Restracomp Postdoctoral Fellowship | Andrew Michael Sydor |

The funders had no role in study design, data collection and interpretation, or the decision to submit the work for publication.

## Author contributions

Andrew Michael Sydor, Data curation, Formal analysis, Supervision, Validation, Investigation, Visualization, Methodology, Writing—original draft, Writing—review and editing; Etienne Coyaud, Data curation, Formal analysis, Investigation, Visualization; Cristina Rovelli, Helen Liu, Validation, Investigation; Estelle Laurent, Investigation; Brian Raught, Conceptualization, Formal analysis, Supervision; Vito Mennella, Conceptualization, Formal analysis, Supervision, Funding acquisition, Visualization, Methodology, Writing—original draft, Project administration, Writing—review and editing

## Author ORCIDs

Andrew Michael Sydor (iD) http://orcid.org/0000-0003-3585-0446
Cristina Rovelli (iD) http://orcid.org/0000-0003-3171-6696
Vito Mennella (iD) http://orcid.org/0000-0002-4842-9012

## Decision letter and Author response

Decision letter https://doi.org/10.7554/eLife.37846.041
Author response https://doi.org/10.7554/eLife.37846.042

# Additional files

## Supplementary files

• Supplementary file 1. Raw BioID and Immunoprecipitation Data. Compilation of all BioID and immunoprecipitation data for all BirA*-tagged constructs used in this study.
DOI: https://doi.org/10.7554/eLife.37846.035

• Supplementary file 2. Primers used in this study. Unless otherwise noted, all primers were used as a part of a Gibson Assembly based cloning strategy.
DOI: https://doi.org/10.7554/eLife.37846.036

• Supplementary file 3. Sequences and manufacture's ID for siRNAs used in this study. All siRNAs were from Ambion (by Life Technologies) except for RTTN that was from Thermo/Invitrogen. Upper case letters represent bases that are present in the target's mRNA sequence.
DOI: https://doi.org/10.7554/eLife.37846.037

• Supplementary file 4. Summary of all statistics used in this study. Corresponding figure numbers are indicated. All statistics in this table were conducted using Barnard's Exact Test, unless otherwise noted.
DOI: https://doi.org/10.7554/eLife.37846.038

• Transparent reporting form
DOI: https://doi.org/10.7554/eLife.37846.039

## Data availability

All data generated or analyzed during this study are included in the manuscript and supporting files.

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
