## [Decision Letter]

Thank you for submitting your article "PPP1R35 is a Novel Centrosomal Protein that Regulates Centriole Length In Concert with the Microcephaly Protein RTTN" for consideration by *eLife*. Your article has been reviewed by three peer reviewers, including Yukiko M Yamashita as the Reviewing Editor and Reviewer #1 and the evaluation has been overseen by Anna Akhmanova as the Senior Editor.

The reviewers have discussed the reviews with one another and the Reviewing Editor has drafted this decision to help you prepare a revised submission.

Summary:

The authors identified PPP1R35 as a novel regulator for centriole formation. They demonstrated that PPP1R35 is a resident centriole protein using live-cell imaging. Using 3DSIM, the authors localize PPP1R35 to an under-characterized portion of the centriole: the inner lumen, immediately distal to the cartwheel. Functional analysis with specific siRNAs against the ORF and 3' UTR revealed that PPP1R35 reduction of function led to a moderate decrease in the number of CEP152 spots and consistently defects in recruitment of components for centriole elongation such as CEP295, SPICE, CPAP and RTTN. Furthermore, BioID and co-IP experiments provided evidence for interactions of PPP1R35 with known centrosomal proteins especially with a significant hit of RTTN, a recently-characterized protein for centriole elongation. Indeed, their localization at centrioles is mutually dependent each other with RTTN having a more significant impact for the PPP1R35 centriolar localization, suggesting that RTTN may be upstream of PPP1R35. Overall, this work will significantly impact the field through the characterization of a novel centriole protein which exists in the centriolar lumen and is essential for centriole growth, two poorly understood aspects of centriole biology. Additionally, this work further details and maps the proteins within distinct regions of the centriole along the proximal-distal axis.

We would like to ask the following issues to be addressed in revision.

Essential revisions:

1) The reviewers are concerned that PPP1R35's localization is assessed only by overexpression of a GFP tagged version, leaving the possibility that this localization might not reflect endogenous one. To address this issue, the authors should (1) use antibody (if a validated one is readily (e.g. commercially) available, but we do not ask to try all possible commercially available ones); (2) if the localization study of the endogenous protein cannot be examined, then the authors should characterize centriole morphology etc. whether or not the overexpression of GFP construct does not interfere centriole/centrosome biogenesis.

2) The efficiency of siRNA knockdown is somewhat unclear (Figure 3—figure supplement 1 only shows siRNA efficiency using cells expressing GFP-tagged protein). We don't know the endogenous level, and how it compares to the expression level of GFP-tagged version. And accordingly, we don't know how much endogenous protein is depleted upon siRNA. Thus, we would like to see a bit more of validation of the phenotype and/or siRNA efficiency. One can try RT-qPCR to see the knockdown efficiency (using wild type cells + siRNA).

---

## [Author Response]

Essential revisions:1) The reviewers are concerned that PPP1R35's localization is assessed only by overexpression of a GFP tagged version, leaving the possibility that this localization might not reflect endogenous one. To address this issue, the authors should (1) use antibody (if a validated one is readily (e.g. commercially) available, but we do not ask to try all possible commercially available ones); (2) if the localization study of the endogenous protein cannot be examined, then the authors should characterize centriole morphology etc. whether or not the overexpression of GFP construct does not interfere centriole/centrosome biogenesis.

We tested six commercially available antibodies in localization experiments. One antibody (ProteinTech 24214-1-AP) generated consistent results to demonstrate co-localization at the centrosome with centriolar markers despite high cytoplasmic and punctate background. To further validate that GFP-PPP1R35 localization reflects the endogenous one, we also examined centrosome morphology of WT and GFP-PPP1R35-expressing U2OS cells labeled with various centrosomal markers by 3DSIM microscopy (see Figure 1—figure supplement 1). In these experiments, we did not observe centriolar defects in the U2OS GFP-PPP1R35 expressing line. Furthermore, in the same line we did not observe defects in centrosome biogenesis relative to WT cells, when comparing the distribution of cells with 0, 1, 2 or > 2 CEP152-labeled centrosomes, since nearly identical distributions of centrosome number were observed. These observations, taken together with the ability of the GFP-PPP1R35 construct to rescue PPP1R35 siRNA knock-down, strongly suggest that this construct is localizing similarly to the endogenous protein and does not impact centrosome structure or biogenesis.

To specifically address this point in the manuscript, we added new panels to Figure 1 (Figure 1D and E) and a new supplemental figure (Figure 1—figure supplement 1), which includes micrographs comparing WT and GFP-PPP1R35-expressing U2OS cells and a graph comparing the distribution of centrosome number in the two cell lines. Furthermore, we added the following to subsection “PPP1R35 is stably associated with the centrosome throughout the cell cycle”:

“To confirm that GFP-PPP1R35 localization is consistent with the endogenous protein, we imaged U2OS cells, stained with antibodies against PPP1R35 and γ-tubulin, by confocal microscopy (Figure 1D) and cells labeled with antibodies against PPP1R35 and CETN1 by 3D structured illumination microscopy (3DSIM), and observed proximity or co-localization (Figure 1E). […] In addition, the construct rescues the centriole duplication phenotype when PPP1R35 levels are knocked down by siRNA (see below; Figure 3C).”

2) The efficiency of siRNA knockdown is somewhat unclear (Figure 3—figure supplement 1 only shows siRNA efficiency using cells expressing GFP-tagged protein). We don't know the endogenous level, and how it compares to the expression level of GFP-tagged version. And accordingly, we don't know how much endogenous protein is depleted upon siRNA. Thus, we would like to see a bit more of validation of the phenotype and/or siRNA efficiency. One can try RT-qPCR to see the knockdown efficiency (using wild type cells + siRNA).

To verify the efficiency of the RNAi knockdown, we conducted RT-qPCR experiments using two sets of primers specific for *c7orf47* (PPP1R35). Using both primer sets, we observed a pronounced knockdown of the endogenous mRNA. We updated the text subsection “PPP1R35 is critical for centriole component recruitment” to reflect this data, which is included as a new figure (Figure 3—figure supplement 1). The Materials and methods section was also updated with the corresponding experimental methodology and the sequences of the primers used are now included in Supplementary file 2.